# Retrieving Instantaneous Extinction of Aerosol Undetected by CALIPSO Layer Detection Algorithm

Feiyue Mao[1,2], Ruixing Shi[1], Daniel Rosenfeld[3], Zengxin Pan[3,*], Lin Zang[4], Yannian Zhu[5,6], Xin Lu[2]

[1]School of Remote Sensing and Information Engineering, Wuhan University, Wuhan 430079, China

[2]State Key Laboratory of Information Engineering in Surveying, Mapping, and Remote Sensing, Wuhan University, Wuhan 430079, China

[3]Institute of Earth Sciences, The Hebrew University of Jerusalem, Jerusalem 91904, Israel

[4]Chinese Antarctic Centre of Surveying and Mapping, Wuhan University, Wuhan 430079, China

[5]School of Atmospheric Sciences, Nanjing University, Nanjing 210023, China

[6]Joint International Research Laboratory of Atmospheric and Earth System Sciences & Institute for Climate and Global Change Research, Nanjing University, Nanjing 210023, China

*Correspondence to*: Zengxin Pan (pzx@whu.edu.cn)

**Abstract.** Aerosols significantly affect the Earth-atmosphere energy balance and climate change by acting as cloud condensation nuclei. Particularly, the susceptibility of cloud and precipitation to aerosols are stronger when aerosols are faint,

but tend to be saturated in polluted conditions. However, previous methodologies generally miss these faint aerosols based on instantaneous observations because they are extremely optically thin to be detected and thereby usually un-retrieved. This result in a large underestimation when quantifying aerosol climate impacts. Here, we focus on retrieving and verifying the instantaneous extinction of undetected faint aerosol by the CALIPSO layer detection algorithm globally. With the constraint of Stratospheric Aerosol and Gas Experiment III on the International Space Station (SAGE III/ISS) observations, the lidar

ratios of undetected faint aerosol are estimated with a globally median of 42.2 and 24.5 sr at the stratosphere and the troposphere, respectively. The retrieved extinction of undetected aerosol during nighttime shows good agreement with the independent 12-months SAGE III/ISS product on a one-degree average. The corresponding correlation coefficients and averaged normalized root mean square error are 0.66 and 100.6%, respectively. The minimum retrieved extinction coefficients are extended to $10^{-3}$ km$^{-1}$ and $10^{-4}$ km$^{-1}$ with the uncertainty of 35% and 125% during nighttime, respectively. The CALIPSO

retrieval during daytime has a positive bias and relatively low agreement with SAGE III/ISS due to the low signal-to-noise ratio caused by sunlight. This study is very potential for improving the understanding of aerosol variations and the quantification of aerosol impacts on climate change globally.

# 1 Introduction

Aerosols significantly affect the Earth-atmosphere system through direct and indirect climate radiative forcing (Boucher et al., 2013). Increased aerosol not only perturbs atmospheric radiative balance by directly interacting with solar radiation (direct effect), but also affects the cloud properties and precipitation by acting as cloud condensation nuclei and ice forming particles (indirect effect) (Dipu et al., 2013; Rosenfeld et al., 2014). Aerosols represent a major uncertainty in global climate change with a low scientific understanding (Schwartz and Andreae, 1996; Lee et al., 2016; Watson-Parris et al., 2020). The Cloud-

Aerosol Lidar and Infrared Pathfinder Satellite Observation (CALIPSO) can detect the vertical properties of aerosols and clouds globally beyond the limitation of passive observation (Winker et al., 2010), providing unprecedented opportunities to advance the understanding of 3D aerosol distribution characteristics and its global climate forcing (Adams et al., 2012; Lu et al., 2018; Song et al., 2021; Winker et al., 2013).

Aerosols are mostly concentrated in the planetary boundary layer (PBL), where optically thick aerosol layers occur and can

usually be detected by the CALIPSO detection algorithm (Li et al., 2017; Kim et al., 2021; Guo et al., 2016). In addition, the persistent faint aerosol in the troposphere and stratosphere has long been considered to have an important effect on the Earth's climate (Turco et al., 1980; Deshler, 2008; Thorsen and Fu, 2015). However, these faint aerosols are usually extremely optically thin to be detected by the CALIPSO layer detection algorithm with a minimum 0.05 threshold of column aerosol optical depth (AOD) (Winker et al., 2013). A previous study indicated the retrieved AODs of aerosols undetected by the CALIPSO layer

detection algorithm can reach 0.03–0.05, which account for approximately 20% of the total AOD and are very important for climatology (Toth et al., 2018; Smirnov et al., 2011; Levy et al., 2013). Thorsen and Fu (2015) pointed out that CALIPSO may have underestimated the magnitude of the aerosol direct radiative effect by 30%–50% due to the undetected faint aerosols by the current CALIPSO algorithm. In addition, aerosol significantly affects cloud formation by acting as cloud condensation nuclei. However, clouds interact directly with ambient sub-cloud aerosol instead of near-surface heavy aerosol, the properties

of which could be very different, especially for aerosol and ice cloud interactions (Rosenfeld et al., 2014). Thus, the ignorance of faint aerosols surrounding high-altitude clouds causes large uncertainty in quantifying the climate effect of aerosols.

Few studies focus on retrieving aerosols undetected by the CALIPSO detection algorithm (Kar et al., 2019; Kim et al., 2017). Kim et al. (2017) attempted to calculate the missing AOD of these undetected aerosols by constraining of the MODIS AOD over ocean. However, that study mainly focused on the AOD of the undetected aerosol with a fixed lidar ratio, but the extinction

of the undetected aerosol was rarely discussed and verified. Additionally, Kim et al. (2017) provided the same lidar ratio (28.75 sr) for the troposphere and stratosphere globally, potentially introducing large uncertainty for extinction retrieval. Recently, the CALIPSO Level 3 Stratospheric Aerosol Profile product was released. However, the purpose of CALIPSO Level 3 products is to provide monthly grid data (5°×20° in latitude and longitude) (Kar et al., 2019), which are insufficient to support studies sensitive to temporal and spatial variations of aerosols, such as studies of aerosol and cloud interactions (Ma et al., 2015).

Furthermore, many studies suggest that CALIPSO may potentially obtain more information on faint aerosols with appropriate data processing (Thomason et al., 2007; Vernier et al., 2009; Kar et al., 2019).

Thus, the present study focuses on retrieving the instantaneous extinction of aerosol undetected by the CALIPSO detection algorithm based on the single-track CALIPSO data. The global distribution of the lidar ratio is obtained with the constraint of the Stratospheric Aerosol and Gas Experiment III on the International Space Station (SAGE III/ISS) observation in the troposphere and stratosphere, respectively. Furthermore, the CALIPSO nighttime and daytime extinction coefficients are retrieved and compared against independent SAGE III/ISS data and CALIPSO Levels 2 and 3 aerosol products. Finally, the impacts of the retrieved lidar ratio and empirical lidar ratio are discussed.

## 2 Data and methodology

### 2.1 CALIPSO data and pre-processing

The CALIPSO mission introduced new technology for retrieving aerosol profiles from space since April 2006, with a dual-wavelength backscattering lidar as the primary payload (Winker et al., 2010). The CALIPSO team has released different levels of products for different scientific objectives. Level 1 products are calibrated observations containing environmental parameters. Level 2 products are physical, chemical, and optical parameters of aerosol layers and cloud layers obtained according to a series of technical routes. The aerosol and cloud layers are firstly detected by the Selective Iterative Boundary Locator (SIBYL) (Vaughan et al., 2009), then classified by the Scene Classification Algorithms (SCA) (Kim et al., 2018), and finally the extinction coefficient is retrieved according to the Hybrid Extinction Retrieval Algorithm (Winker et al., 2010; Young et al., 2018). Level 3 products provide monthly averaged gridded global distribution data of clouds and aerosols (Kar et al., 2019).

This study uses CALIPSO Level 1B for the extinction retrieval of the undetected aerosol by SIBYL from June 2017 to May 2020 (Table 1). In addition, the CALIPSO Level 3 monthly-averaged Stratospheric Aerosol Profile product with a resolution of 5°×20° in latitude and longitude is compared with the retrieved extinction of undetected aerosol. To improve the signal-to-noise ratio (SNR) and avoid contamination by clouds and detected aerosols, the CALIPSO Level 1B total attenuated backscatter (TAB) data were pre-processed according to the following steps:

(1) We removed the affected CALIPSO observations according to Low Laser Energy Technical Advisory due to the effects of an elevated frequency of low-energy laser shots of CALIPSO within the South Atlantic Anomaly (SAA) (https://www-calipso.larc.nasa.gov/resources/calipso_users_guide/advisory).

(2) The clouds and aerosol layer detected by the SIBYL and the data below them were removed. We used a threshold value of 0.5 in the attenuated color ratio (the ratio of the TAB at 1064 and 532 nm) to remove undetected tenuous cirrus clouds, similar to the data screening method of the CALIPSO Level 3 Stratospheric Aerosol Profile product (Kar et al., 2019).

(3) The vertical resolution of the CALIPSO Level 1B TAB profiles varies with the height of 30, 60, 120, and 300 m for −0.5–8, 8–20.2, 20.2–30.1, and 30.1–40 km, respectively. Referring to Kim et al. (2017), the TAB profiles are reduced to a vertical resolution of 300 m by linear interpolation to improve the SNR, followed by a vertical moving mean filtering (with a 5-point window) and horizontal averaging to 20 km to retrieve the extinction of undetected aerosol.

**Table 1.** Data used in the study with their sources and parameters.

| Source | Product | Parameter |
|---|---|---|
| CALIPSO | Level 1B Profile, Version 4.10 | Total attenuated backscatter at 532 nm, tropopause height, molecular number density, ozone number density |
| | Level 2 Vertical Feature Mask, Version 4.20 | Feature classification flag of aerosol and cloud |
| | Level 3 Stratospheric Aerosol Profile Monthly Product | Aerosol extinction coefficient with background mode |
| SAGE III/ISS | Level 2 Solar Event Species Profiles, V051, 0.5 km vertical interval | Aerosol extinction coefficient at 521, 450, 755 nm and 1022 nm |

## 2.2 SAGE III/ISS data and pre-processing

The Stratospheric Aerosol and Gas Experiment (SAGE) was developed to obtain vertical profiles of aerosol optical properties since 1984. SAGE could detect the extinction of faint aerosol in the upper troposphere and the stratosphere (Mauldin III et al., 1985; Damadeo et al., 2013). SAGE III conducts solar and lunar occultation measurements globally while orbiting the Earth on the International Space Station (ISS). Light passes through the atmosphere and is attenuated by some combination of scattering and absorption of molecules, particles, and clouds. The extinction coefficients are then derived based on the recorded spectra (Cisewski et al., 2014; Thomason et al., 2010). The aerosol extinction of the SAGE III/ISS product with a vertical resolution of 0.5 km at the 521 nm channel, which is closest to the CALIPSO 532 nm channel, is used for comparison and validation in this study. Only the SAGE III/ISS solar occultation product was used in this study because of the absence of aerosol extinction information in the SAGE III/ISS lunar occultation product (Table 1).

A low bias in the extinction coefficients of the SAGE III/ISS aerosol product is observed at 521 nm due to the ozone interference in the retrieval algorithm. This finding is more pronounced at mid-latitudes and altitudes between 20 and 25 km (Knepp et al., 2021). The following equation is therefore used to correct the extinction at 521 nm (Knepp et al., 2021):

$$log\ \sigma_{521} = \frac{log\ (\frac{\sigma_{450}}{\sigma_{755}}) \times log\ (\frac{521}{755})}{log\ (\frac{450}{755})} + log\ (\sigma_{755}), \qquad (1)$$

where $\sigma$ is the extinction coefficient, and the numbers represent the wavelength. We removed the bins in the SAGE III/ISS aerosol extinction profile with color ratio (the ratio of the aerosol extinction at 521 and 1022 nm) in the range of 0.8 to 1.2 to avoid cloud contamination (Schoeberl et al., 2021).

## 2.3 Match of CALIPSO and SAGE III/ISS

Since only daytime data from SAGE III/ISS are available, the CALIPSO orbits are spatially and temporally matched to the nearest SAGE III/ISS observations on the same calendar date with the consideration for a  smaller temporal-spatial variation of faint aerosol comparing strong aerosol at the near-surface. The horizontal resolution of SAGE III/ISS occultation observations is low with ~300 km (https://space.oscar.wmo.int/instruments/view/sage_iii). Thus, we selected a 2° × 1°

(longitude × latitude) grid centered on the SAGE III/ISS observations to match CALIPSO instantaneous observation. To ensure enough CALIPSO profiles are included for each successfully matched sample, the CALIPSO track crossed the grid and have to exceed 0.75° latitude (Figure 1a). The 2° longitude is to obtain the successfully matched samples as soon as possible. Figure

1b shows the global distribution of nighttime CALIPSO and SAGE III/ISS match numbers in 20°×20° grids for three years from June 2017 to May 2020. No successful match in the grids is found in the black boundary due to the removal of low-energy laser shots of CALIPSO in the SAA region. Finally, 1349 and 1325 profiles are successfully matched for CALIPSO nighttime and daytime data with SAGE, respectively.

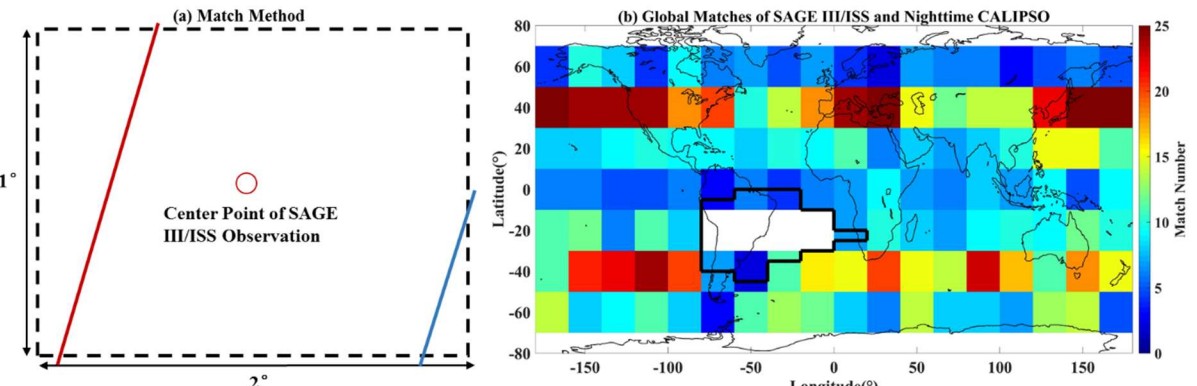

**Figure 1. (a) Schematic of the CALIPSO match to the SAGE III/ISS. The red circle represents the center point of SAGE III/ISS observations. The red and blue lines represent cases of successful and failed (CALIPSO track is less than 0.75° in the grid) matches, respectively. (b) Global matches of nighttime CALIPSO and SAGE III/ISS from June 2017 to May 2020. The black boundary represents the South Atlantic Anomaly (SAA), where CALIPSO is experiencing an elevated frequency of low-energy laser shots. The color bar represents the number of matches in each 20°×20° grid.**

**2.4 Retrieving instantaneous extinction of undetected aerosol under the constraint of SAGE III/ISS**

In this study, the undetected aerosol extinction coefficient is retrieved by the Fernald method, similar to CALIPSO Level 2 and Level 3 aerosol products (Young and Vaughan, 2009; Kar et al., 2019; Young et al., 2018). Based on the pre-processed TAB (i.e., $\beta'(r)$), the particulate backscatter coefficient (i.e., $\beta_p(r)$) is solved by iterating Eqs. (2) and (3c) in the following equations:

$$\beta_p(r) = \frac{\beta'(r)}{T_m^2(r)T_{o_3}^2(r)T_p^2(r)} - \beta_m(r), \tag{2}$$

$$T_m^2(r) = exp\left(-2\int_0^r \alpha_m(r')\, dr'\right), \tag{3a}$$

$$T_{o_3}^2(r) = exp\left(-2\int_0^r \alpha_{o_3}(r')\, dr'\right), \tag{3b}$$

$$T_p^2(r) = exp\left(-2\eta_p S_p \int_0^r \beta_p(r')\, dr'\right), \tag{3c}$$

$$\alpha_p(r) = S_p(r)\beta_p(r), \tag{4}$$

where $T_m^2(r)$, $T_{o_3}^2(r)$, and $T_p^2(r)$ represent the molecular, ozone, and particulate two-way transmittances, respectively. The molecular backscatter coefficients ($\beta_m(r)$) and molecular and ozone two-way transmittances ($T_m^2(r)$ and $T_{o_3}^2(r)$) can be calculated from the molecular number density and ozone number density provided by CALIPSO Level 1B product, respectively. The $\alpha_m(r)$, $\alpha_{o_3}(r)$ and $\alpha_p(r)$ represent the extinction coefficient of molecular, ozone, and particulate, respectively. The retrieval algorithm has several basic settings. The multiple scattering coefficient ($\eta_p$) for undetected aerosol

particles is set to 1 as considered in the retrieval of the CALIPSO Level 2 product (Young et al., 2018). Meanwhile, the bin at 36 km is considered aerosol-free (i.e., $\beta_p(0) = 0$, $T_p^2(0) = 1$) (Kar et al., 2019).

When using the Fernald method to retrieve aerosol extinction coefficients, the lidar ratio ($S_p(r)$) is a key parameter (Fernald, 1984; Fernald et al., 1972), which is often set based on aerosol type or empirical values (Young et al., 2018; Kar et al., 2019). The backscattered signal of undetected aerosols is extremely weak to be detected and classified by the CALIPSO layer

detection and classification algorithms (Kim et al., 2017; Toth et al., 2018). The extinction retrieval of undetected aerosols is very sensitive to the lidar ratio (Kim et al., 2017). Therefore, to obtain the appropriate lidar ratio of undetected aerosol, we retrieve the lidar ratio by using SAGE III/ISS 521 nm AOD as a constraint, and the algorithm flow is shown in Figure 2.

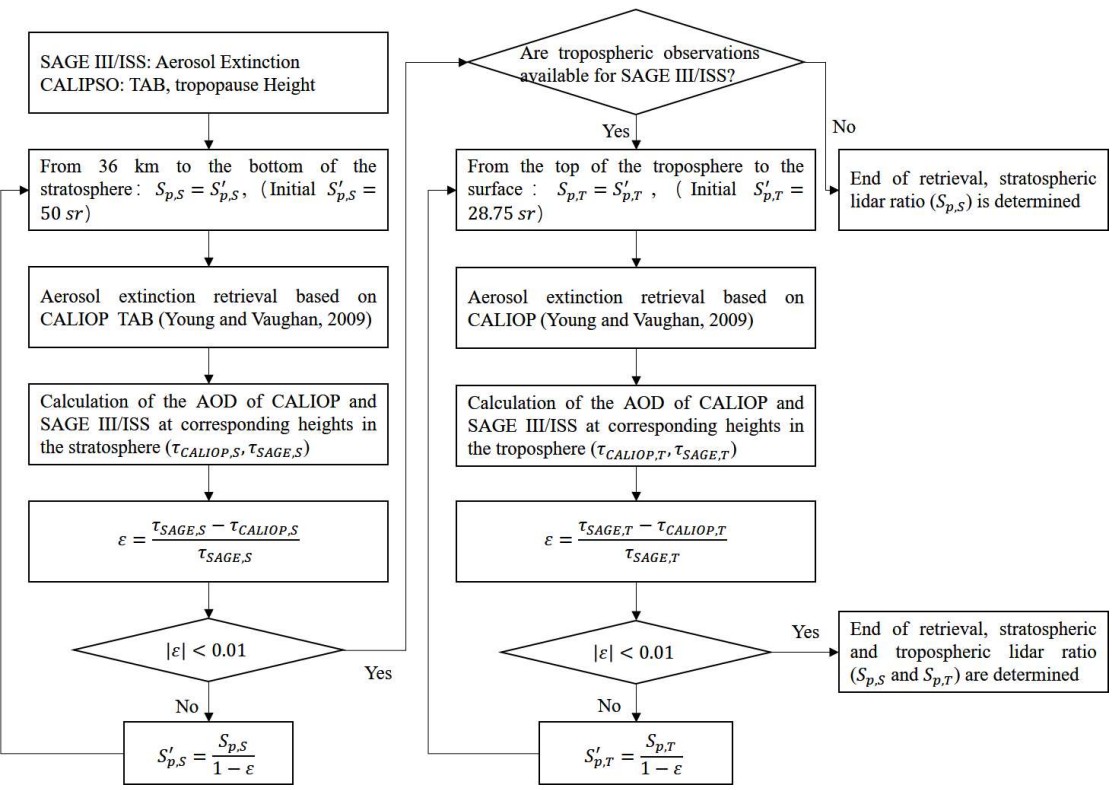

**Figure 2. Flowchart for the retrieval of lidar ratio by using SAGE III/ISS AOD as a constraint.**

We perform the retrieval of the lidar ratio separately because the aerosol compositions in the troposphere and stratosphere are different. For the stratosphere, the initial lidar ratio ($S_{p,S}$) is set to 50 sr, which is widely assumed for stratosphere aerosol (Kar

et al., 2019; Sakai et al., 2016; Khaykin et al., 2017), and the extinction retrieval is performed from 36 km to the bottom of the stratosphere. The AOD of CALIPSO and SAGE III/ISS ($\tau_{CALIOP,S}$ and $\tau_{SAGE,S}$) for the same altitude bins in the stratosphere and the deviation ($\varepsilon$) between them are calculated. The lidar ratio is iteratively modified and the extinction and AOD of

CALIPSO are recalculated until $|\varepsilon| < 0.01$. The same procedure is performed in the troposphere; the difference between the retrieval altitude and using an initial lidar ratio ($S_{p,T}$) of 28.75 sr refers to the estimate by Kim et al. (2017).

The tropospheric and stratospheric lidar ratios are retrieved globally based on matched SAGE III/ISS and CALIPSO profiles and counted at each $20°\times20°$ grid. When performing the extinction retrieval of CALIPSO, $S_{p,S}$ and $S_{p,T}$ can be selected depending on which grid the profile is located on. The constrained retrieval of the lidar ratio uses nighttime CALIPSO and

daytime SAGE III/ISS profiles given that daytime CALIPSO observations are affected by solar background noise and have a much lower SNR than nighttime observations (Hunt et al., 2009). The implicit assumption is that diurnal variations in undetected aerosols are ignored. To obtain a consistent lidar ratio retrieval dataset and validation dataset, we used data from the first two months of each quarter to derive the lidar ratio and those of the last month for validation. Thus, for three years from June 2017 to May 2020, 24 months of data are retrieved to determine the lidar ratio and 12 months of data for validation.

For the retrieved extinction of undetected aerosol, we calculated the uncertainty to assess the reliability of the results according to the algorithm of CALIPSO Level 2 aerosol product (Young et al., 2013), where the main equations are as follows:

$$\frac{\Delta\beta'_N(r)}{\beta'_N(r)} = \left\{\left[\frac{\Delta\beta'(0,r)}{\beta'(0,r)}\right]^2 + \left[\frac{\Delta C_N(r_N)}{C_N(r_N)}\right]^2\right\}^{1/2}, \tag{5}$$

$$\left(\Delta\beta_p(r)\right)^2 = \beta_T^2(r)\left[\left(\frac{\Delta\beta'_N(r)}{\beta'_N(r)}\right)^2 + \left(\frac{\Delta T_M^2(r_N,r)}{T_M^2(r_N,r)}\right)^2 + \left(\frac{\Delta T_P^2(r_N,r)}{T_P^2(r_N,r)}\right)^2\right] + \left(\Delta\beta_M(r)\right)^2, \tag{6}$$

$$\Delta\sigma_P(r) = \left[(\frac{\Delta S_p}{S_p})^2 + (\frac{\Delta\beta_p(r)}{\beta_p(r)})^2\right]^{1/2}\sigma_P(r), \tag{7}$$

where $\Delta\beta_p(r)$ and $\Delta\sigma_P(r)$ in Eqs (6) and (7) are the particle backscatter uncertainty and particle extinction uncertainty, respectively; they are the target parameters for the calculation. Eq. (5) is the formula for one of the terms of Eq. (6), where $\Delta\beta'_N(r)$ is the uncertainty of the renormalized TAB, $\Delta\beta'(0,r)$ is the uncertainty of the TAB, and $\Delta C_N(r_N)$ is the uncertainty of renormalization. The error due to renormalization is negligible (Kim et al., 2017) because the starting altitude of retrieval ($r_N$ =36 km) is consistent with the calibration region (36–39 km) for the CALIPSO Level 1B Version 4 product (Kar et al.,

2018); therefore, $\Delta C_N(r_N)$ is set to 0. The standard deviation of the TAB is used to approximate $\Delta\beta'(0,r)$ because the TAB in this study was pre-processed.

Uncertainty is found in the calibration factor in $\Delta\beta'(0,r)$, which contains systematic and random components (Young et al., 2013), and this approximation neglects the systematic error in the calibration factor, producing a low bias in the uncertainty calculation. Fortunately, the calibration factor bias of the nighttime CALIPSO V4 product has been reduced to 1.6%±2.4%

(Kar et al., 2018). Additionally, Kim et al. (2017) pointed out that the bias caused by the lidar ratio is dominated in the retrieval. Thus, we consider ignoring the calibration factor in the systematic error. The other terms in Eq. (6), total backscatter coefficient ($\beta_T(r)$), molecular and particle two-way transmittance uncertainty ($\Delta T_M^2(r_N,r)$ and $\Delta T_P^2(r_N,r)$), and molecular backscatter

uncertainty ($\Delta\beta_M(r)$) are calculated in the same way as in Young et al. (2013) and are not repeated here. $S_p$ and $\Delta S_p$ in Eq. (7) are selected from the median and median absolute deviation, respectively, in the retrieved 20°×20° grid lidar ratio based on CALIPSO profile locations.

## 3 Results and analysis

### 3.1 Global gridded distribution of lidar ratio

Figure 3 shows the global distribution of the median lidar ratios in 20°×20° grids retrieved by CALIPSO under the SAGE III/ISS 521 nm products constraint. The median of the global stratospheric lidar ratio is 42.2 sr, whereas the lidar ratio is smaller at high latitudes than that near the equator (Figure 3a), which is consistent with the latitude-lidar ratio distribution in Kar et al. (2019). The median global tropospheric lidar ratio is smaller (24.5 sr) and shows a different trend from that of the stratosphere, slightly decreasing from the northern to the southern hemisphere (Figure 3b). In the following, we retrieve the extinction of CALIPSO undetected aerosol with the median lidar ratios of the stratosphere and troposphere in the grid, where the CALIPSO profile is located on. In addition, the median absolute deviation of the lidar ratio in the grid is used to calculate the uncertainty of the extinction (Eq. (7)).

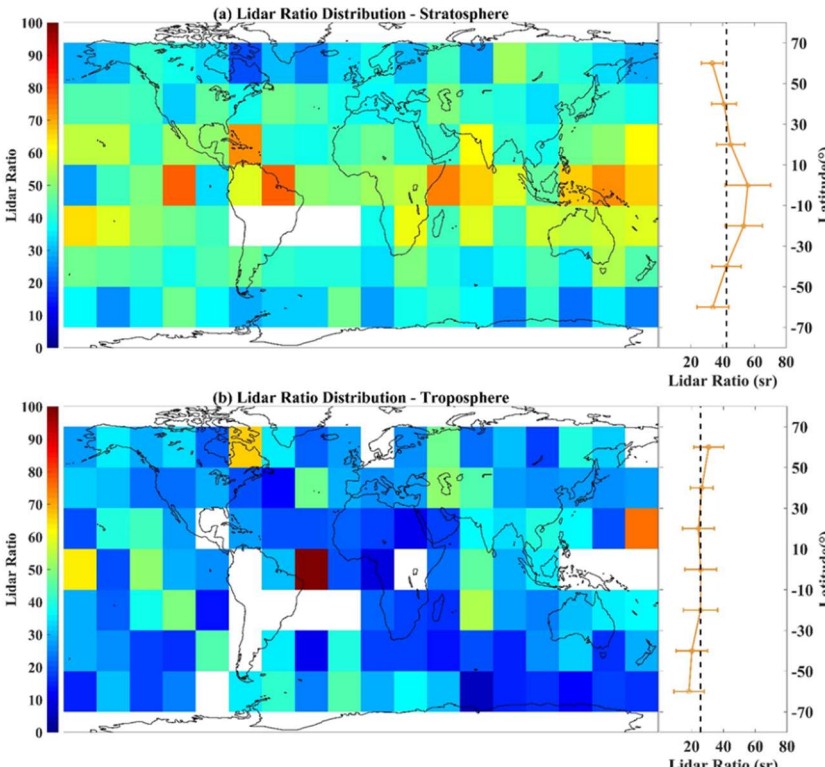

**Figure 3. (a) Global stratospheric distribution of lidar ratios with a grid size of 20° x 20°. The color bar represents the lidar ratio value. The line on the right shows the median variation at 20° intervals from −70° to 70° (latitude) globally, and the error bar represents the median absolute deviation. (b) Same as (a), but for the troposphere. A blank grid indicates that no data is available.**

### 3.2 Comparison with SAGE III/ISS Aerosol Product

Figure 4a shows a case of the retrieved CALIPSO extinction at latitude 33° on August 26, 2019. An undetected faint aerosol layer (extinction coefficients around 0.005 km$^{-1}$) is connected to the detected stratospheric aerosol layer provided by the CALIPSO Level 2 aerosol product at altitudes of 15 km to 20 km around 10°N to 40°N latitude. Figure 4b shows high-consistent extinctions of CALIPSO undetected aerosol and matched SAGE III/ISS 521 nm aerosol product (red dash line in Figure 4a) above 15 km. Additionally, this profile comparison demonstrates the feasibility of ignoring the diurnal variation of undetected aerosols.

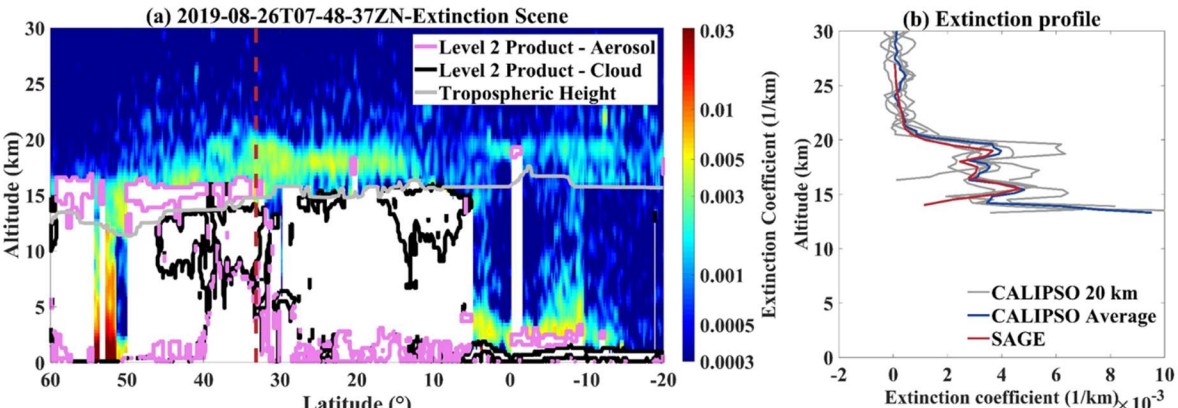

**Figure 4. (a) Latitude-altitude undetected aerosol extinction based on CALIPSO nighttime data on August 26, 2019. The color represents the extinction coefficient (km$^{-1}$). The purple and black boundary lines represent the detected aerosol and cloud layers provided by CALIPSO Level 2 products, respectively. The gray line represents tropospheric height. The red dash line is the observation position of SAGE III/ISS. The white areas represent the removed data inside and below the detected layers. The retrieved faint aerosol at 20 km is shown after additional mean filtering (3×3 window) to highlight the faint aerosol area. (b) Comparison of faint aerosol extinction (km$^{-1}$) profile for matched CALIPSO and SAGE III/ISS 521 nm aerosol product. The gray lines represent the undetected aerosol extinction of CALIPSO retrieval at a resolution of 20 km horizontally and 0.3 km vertically, and the blue line represents averaged gray lines. The red line represents the aerosol extinction from SAGE III/ISS.**

The nighttime CALIPSO undetected aerosol extinction and SAGE III/ISS 521 nm aerosol extinction show good agreement for the 12-month validation dataset (Figure 5a), with the average retrieved aerosol extinction (black line) closing to the 1:1 line. The correlation coefficients (R) and normalized root mean square error (NRMSE) are 0.66 and 100.6% based on the independent 12-month SAGE validation dataset, respectively. The CALIPSO extinction in Figure 5a is the one-degree averaged profile (e.g., red line in Figure 4b) of the CALIPSO 20 km profiles (e.g., gray line in Figure 4b) over the matched range. Therefore, considering the systematic error of the lidar ratio in Eq. 7, we calculate the uncertainty according to the following equation:

$$(\Delta\alpha_{1°})^2 = \sum_{i=1}^{n}(\frac{1}{n} \times \left(\frac{\Delta\beta_p}{\beta_p}\right) \times \alpha_{20\,km})^2 + \left(\sum_{i=1}^{n}(\frac{1}{n} \times \left(\frac{\Delta S_p}{S_p}\right) \times \alpha_{20\,km})\right)^2,$$ (8)

where $n$ represents the number of CALIPSO 20 km profiles in the matching range, $\alpha_{20\ km}$ is the 20 km aerosol extinction of
CALIPSO, and $\Delta\alpha_{1°}$ is the uncertainty for one-degree aerosol extinction of CALIPSO.

Figure 5b shows the relationship between the extinction and relative uncertainty (ratio of $\Delta\alpha_{1°}$ and $\alpha_{1°}$) of the CALIPSO
retrieval. The relative uncertainty increases as the extinction coefficient decreases because low extinction corresponds to low
particle concentrations and weak backscatter signals, resulting in lower SNR. The averaged black line in Figure 5b show the
mean relative uncertainties of CALIPSO, specifically ~35% and ~125% for the retrieved extinction of $10^{-3}$ and $10^{-4}$ km$^{-1}$,
respectively. This indicates the retrieved extinction of undetected aerosol is much smaller than the low boundary of the detected
aerosol extinction ($10^{-2}$ km$^{-1}$) from the CALIPSO Level 2 extinction product with a 40% uncertainty (Kacenelenbogen et al.,
2011; Toth et al., 2018; Winker et al., 2013; Winker et al., 2009). Similarly, Watson‐Parris et al. (2018) noted through the
model that the minimum value of aerosol extinction at 0–15 km should be close to $10^{-4}$ km$^{-1}$, whereas CALIPSO Level 2
aerosol products remain above $10^{-2}$ km$^{-1}$, and the mean fraction of aerosol undetected by CALIPSO daytime (nighttime)
retrievals is 92% (87%) globally.

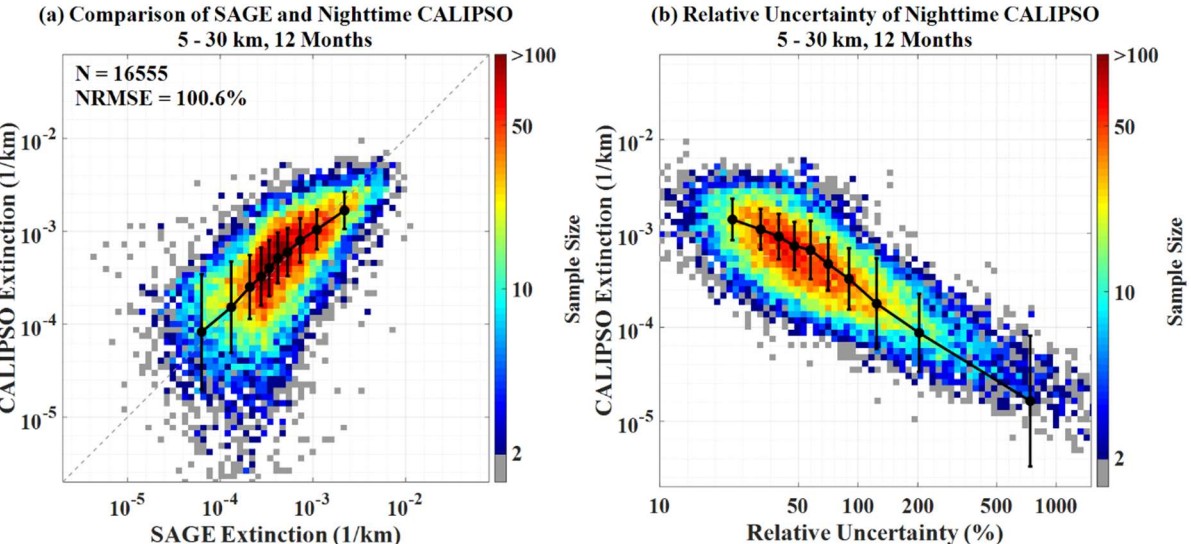

**Figure 5. (a) Correlation plots of the retrieval within the matching grid of CALIPSO nighttime and SAGE III/ISS product from 5 km to 30 km for 12 months of validation. The color bar represents the sample size. The black bins represent the mean values of each 10% quantile (0-10%, 10-20%...and 90-100%) of SAGE III/ISS 521 nm aerosol product and corresponding CALIPSO retrieval.**
**The I-type bars indicate the standard error of each 10% quantile CALIPSO retrieval. (b) The relative uncertainty of one-degree CALIPSO extinction.**

The agreement between daytime CALIPSO retrieval and SAGE III/ISS between 5 km to 30 km (with R=0.25 and
NRMSE=454.5%) is poorer than during nighttime (with R=0.66 and NRMSE=100.6%) (Figures 5a and 6a). The poorer
agreement is due to the lower SNR of CALIPSO, which is attributed to sunlight during the daytime (Figure 6b) (Hunt et al.,
2009). The distribution of lidar signals received by photomultipliers is Neyman type-A (originally defined for a Poisson
process) (Teich, 1981), thereby introducing a positive bias in the extinction retrieval calculation when the SNR is low. Also,
Young et al. (2013) noted that the CALIPSO retrievals with SNR≤1 usually contain a positive bias. The SNR during daytime

above 20 km is usually less than 1 for TAB at a 20 km horizontal scale (Figure 6b), leading to a significantly positive bias in the retrieval (Figure 6a), as noted by Young et al. (2013). In addition, a layer detection algorithm possibly misses more optically

thick aerosol layers at low SNR during daytime compared with that during nighttime (Huang et al., 2015), thereby causing large retrieved aerosol extinctions in Figure 6a.

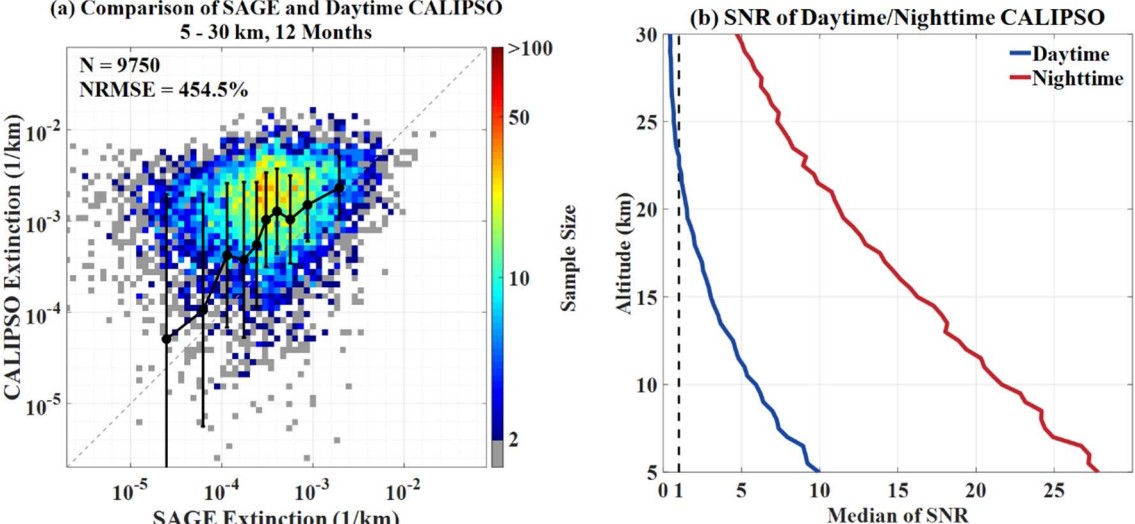

**Figure 6. (a) Same as in Figure 5a, but at daytime. (b) SNR profiles of the TAB at 20 km horizontal resolution at daytime (blue) and nighttime (red). The SNR is calculated according to $SNR = \mu / \sigma$ based on the pre-processed TAB with 20 km horizontal resolution**
**in the matching range, where $\mu$ and $\sigma$ are the mean and the standard deviation of the signal, respectively.**

### 3.3 Comparison with CALIPSO Level 2 and Level 3 Products

Figure 7 shows a case to compare the retrieved undetected aerosol with CALIPSO Level 2 product during a wildfire event in Australia in August 2019. Smoke and dust transmission trajectory are shown in the red dash boxed area of the Terra MODIS true-color image (Figure 7a). In the scene of aerosol extinction coefficient (Figure 7b), CALIPSO Level 2 product only shows
two detected strong aerosol layers (shown as purple boundaries) between −25° and −30° latitude, which is labeled as elevated smoke (Figure 7c). These aerosol layers should belong to a continuous one (shown in the red dashed box), but the Vertical Feature Mask (VFM) does not show the aerosol (~0.01 km⁻¹) between the two strong aerosol layers (~0.03 km⁻¹), which may be below the threshold of the SIBYL. The attenuated scattering ratio (ASR) (Figure 7d), which is the ratio of the CALIPSO Level 1 total attenuated backscatter and attenuated molecular backscatter product, also demonstrates the overall continuous
nature of this aerosol layer.

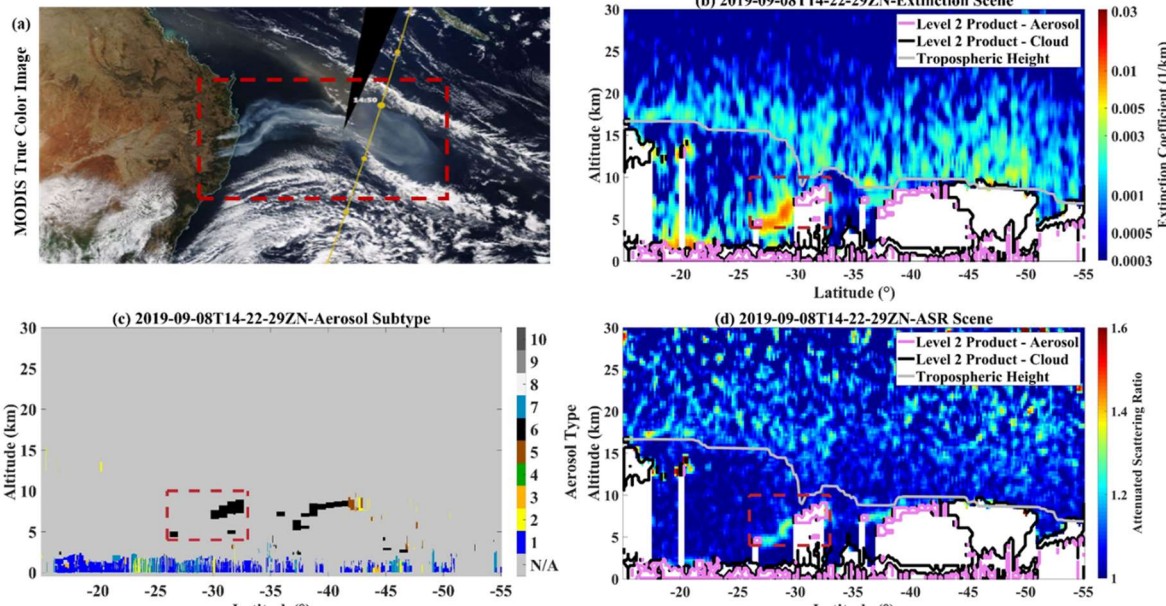

**Figure 7. (a) MODIS Terra true-color image in the daytime and the passing CALIPSO track (yellow line) at night. (b) Latitude–altitude aerosol extinction of the corresponding nighttime CALIPSO track, same as in Figure 4a. The purple and black boundary lines and extinction inside represent the detected aerosol and cloud layers provided by CALIPSO Level 2 products, respectively. (c) Aerosol subtypes in CALIPSO VFM product (N/A=not applicable, 1=marine, 2=dust, 3=polluted continental/smoke, 4=clean continental, 5=polluted dust, 6=elevated smoke, 7=dusty marine, 8=PSC aerosol, 9=volcanic ash, 10=sulfate/other). (d) Attenuated scattering ratio.**

Figures 8a and 8b show the spatial distribution of aerosol extinction averaged in June and August 2019 at 17 km altitude from CALIPSO Level 3 monthly-averaged Stratospheric Aerosol Profile product with the resolution of 5°×20° in latitude and longitude (Kar et al., 2019). A significant amount of aerosol enhancement was observed in the stratosphere in August in the northern hemisphere (Figure 8b), possibly due to the eruption of the Raikoke Volcano in June 2019 (Knepp et al., 2021; Kloss et al., 2021). We selected two CALIPSO tracks across aerosol enhancement areas in June and August (Figures 8c and 8d), respectively. The stratosphere at the northern hemisphere latitudes is clean, whereas natural dust aerosol prevails in the lower troposphere on June 10 when Raikko has not yet erupted (Figures 8c and 8e). The clean condition shown by our retrieval is consistent with the CALIPSO Level 3 products that indicate the clean stratosphere at a monthly temporal scale.

Following the onset of volcanic eruptions, strong stratospheric aerosol layers are found in the stratosphere between 50°N and 60°N that are classified as sulfate by the VFM (Figure 8f). As shown in the red dash box of Figure 8d, aerosol extinction enhancement (~0.005 km$^{-1}$) occurs around 17 km near 40°N to 5°N, which corresponds to the monthly average scale aerosol contamination in the stratosphere throughout the Northern Hemisphere in Figure 8b, but is not captured by CALIPSO Level 2 products (Figure 8f). Therefore, the retrieved undetected aerosol extinction can well capture the aerosol enhancement from special events at a horizontal resolution of 20 km (Figure 8d). The color ratios, particle depolarization ratios, and integrated attenuated backscatter are extracted manually for the red dashed region (16 km to 20 km, 40°N to 5°N) with an average of 0.17, 0.02 and 0.00033 sr$^{-1}$, respectively. Using these optical and non-optical properties (i.e., center height, temperature and

latitude), aerosol subtypes can be determined by the CALIPSO Scene Classification Algorithms (Kim et al., 2018). The results

show that the aerosol subtype in this region is sulfate, which supports that the aerosol enhancement is more likely to be from

the eruption of the Raikoke Volcano.

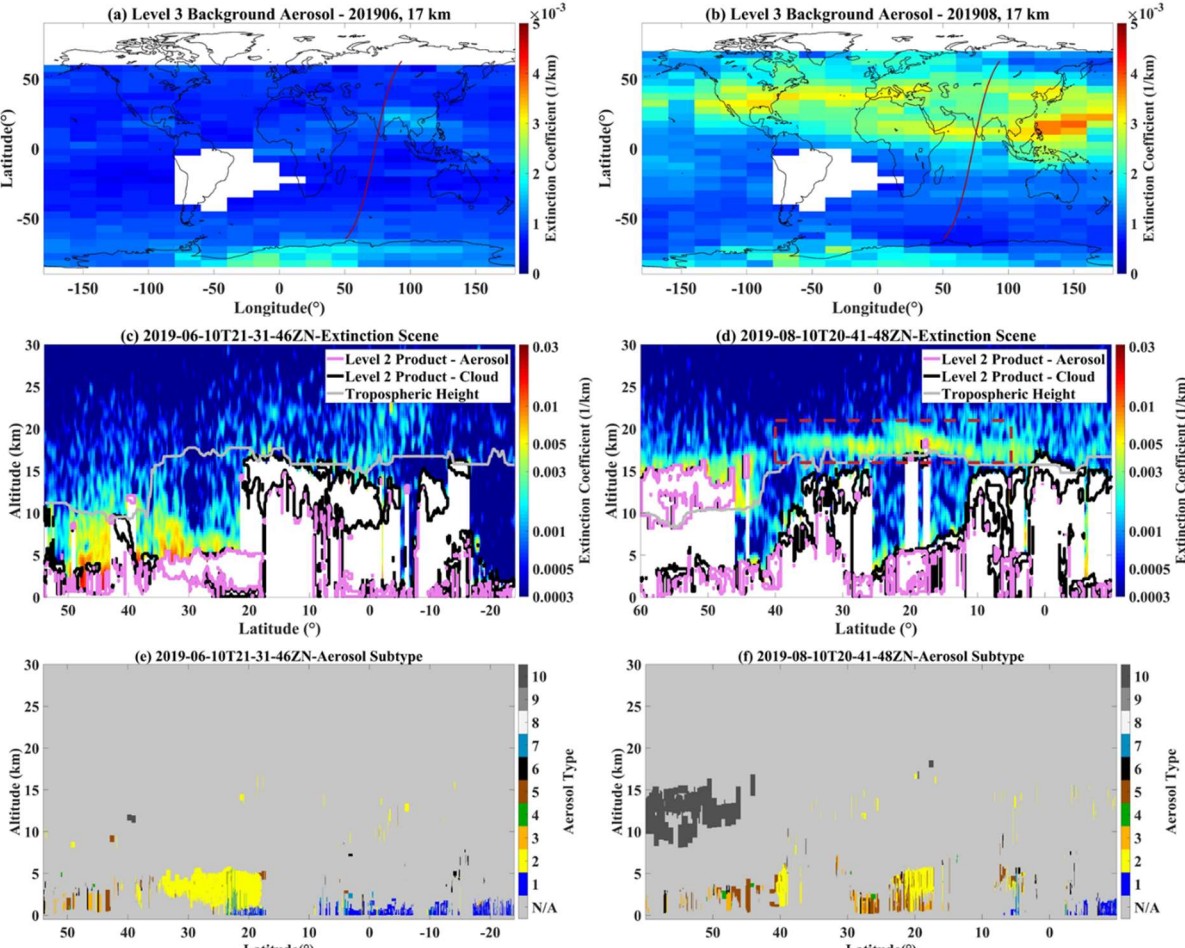

**Figure 8. (a) and (b) are the stratospheric extinction distributions of CALIPSO Level 3 Stratospheric Aerosol Profile products at 17 km in June and August, respectively. (c) and (d) are the retrieved aerosol extinction scenes based on CALIPSO instantaneous data**

**on June 10 and August 10, respectively, consistent with Figure 4a. The corresponding trajectories for the two scenes are shown as red lines in (a) and (b), and the corresponding aerosol subtypes are shown in (e) and (f), the same as in Figure 7c.**

### 3.4. Discussion on the use of lidar ratio

As mentioned in Section 2.4, the initial stratosphere and troposphere lidar ratios were derived from the empirical value (50 sr) of CALIPSO Level 3 stratospheric aerosol product (Kar et al., 2019) and the lidar ratio (28.75 sr) obtained by Kim et al. (2017),

respectively. The latter is estimated from the retrieved CALIPSO column-integrated extinction with MODIS AOD constraints. As shown in Figure 9, the retrieved extinction using the fixed lidar ratio is higher than that using the SAGE-constrained lidar ratio because the median lidar ratio of the former (50 and 28.75 sr) is larger than the latter (42.2 and 24.5 sr). However, the

NRMSE of retrieved extinction decreased by about 15% (from 120.2% to 105.6%) when changed the fixed lidar ratio to the SAGE-constrained lidar ratio in global. Particularly, when using the fixed lidar ratio of 50 sr in the high latitude stratosphere, it could result in a larger bias because the fixed lidar ratio is more different from the SAGE-constrained lidar ratio (~35 sr) (Figure 3a). Therefore, these indicate a better accuracy of retrieved undetected aerosol extinction using the SAGE-constrained lidar ratio in global.

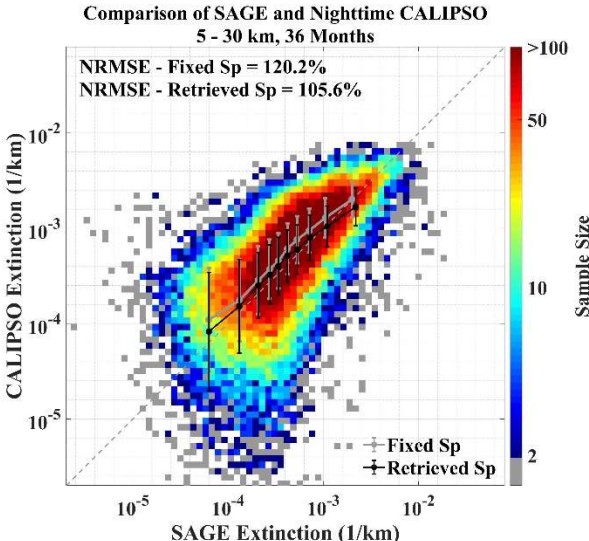

**Figure 9. The colored scatter plot is the same as that in Figure 5a, but the CALIPSO extinction are retrieved using fixed lidar ratios of 50 and 28.75 sr in stratosphere and troposphere from June 2017 to May 2020, respectively. The gray and black lines are the mean value of of each 10% quantile (as in Figure 5a) of the CALIPSO retrieved extinction using the fixed and our retrieved lidar ratios, respectively.**

Figure 7b illustrates a possibly missed smoke from a wildfire. Based on the SAGE-constrained lidar ratio (median 42.2 and 24.5 sr), we retrieved and see the undetected aerosol by CALIPSO Level 2 products, which connects with two strong aerosol layers. The lidar ratio for the smoke reported in the CALIPSO Level 2 Version 4 product is 70±16 sr (Young et al., 2018), which is very different from the SAGE-constrained lidar ratio for the troposphere at this location. Theoretically, a larger lidar ratio will derive a larger extinction in the retrieval. This indicates that the undetected aerosol extinction should be larger if using the smoke lidar ratio of 70±16 sr. However, so for, this bias cannot be avoided here because an automatic classification is impossible when we do not know the boundaries of those aerosols. Therefore, we have to treat the stratospheric (or tropospheric) undetected aerosols as a whole and assign the same lidar ratio regardless of the aerosol type in this study. Although the retrieved extinction in Figure 7 is biased, it demonstrated the importance of retrieving high spatial-temporal resolution undetected aerosol extinction. A solution to reduce this bias is to develop a more effective layer detection and classification algorithm, and our team is already working on it (Mao et al., 2021).

**4 Conclusions**

An abundance of faint aerosols in the background atmosphere significantly affects the global climate (Chazette et al., 1995; Deshler, 2008). However, these faint aerosols are extremely optically thin to be instantaneously detected and rarely retrieved by current methodologies (Watson-Parris et al., 2018; Toth et al., 2018). With the constraint of SAGE aerosol observations in global, this study retrieved instantaneous extinction of aerosol undetected by the CALIPSO layer detection algorithm based on CALIPSO Level 1B data. The main conclusions are summarized as follows:

(1) The lidar ratio for the stratosphere and troposphere in global is derived based on CALIPSO instantaneous observations using SAGE III/ISS AOD as a constraint. The derived lidar ratio is significantly higher in the stratosphere (median 42.2 sr) than that in the troposphere (median 24.5 sr). The derived lidar ratio peak at the equator and decrease with latitude at the stratosphere, while the lidar ratio variations are small at the troposphere in global.

(2) The retrieved undetected aerosol extinction based on CALIPSO nighttime instantaneous observations shows good
agreement with the SAGE III/ISS product on a 1° average. The correlation ($R$) and NRMSE are 0.66 and 100.6% based on the independent 12-months SAGE III/ISS data, respectively. The uncertainties of retrieved extinction coefficients at $10^{-3}$ km$^{-1}$ and $10^{-4}$ km$^{-1}$ are ~35% and 125% during nighttime, respectively.

(3) The comparison of retrieved undetected aerosol extinction based on globally fixed and SAGE-constrained lidar ratios indicates the NRMSE decreased by about 15% (from 120.2% to 105.6%) during nighttime. Additionally, the CALIPSO
retrieval during daytime has a positive bias and relatively low agreement with SAGE III/ISS; it exhibits R and NRMSE of 0.25 and 454.5%, respectively, due to the low signal-to-noise ratio caused by sunlight.

(4) In the case of the Australian wildfire event, instantaneous retrieved extinction of missed aerosol from CALIPSO Level 2 products provides more details of aerosol distribution. In addition, compared with the CALIPSO Level 3 stratospheric aerosol product, the retrievals show consistent aerosol enhancement possibly due to the eruption of Raikoke Volcano, but
at a higher spatial-temporal resolution.

This study allows more efficient capture of aerosol vertical properties of events, such as volcanic eruptions and wildfires, by acquiring instantaneous and high-resolution faint aerosols globally (Vernier et al., 2015; Andersson et al., 2015). Moreover, a large potential for new insights is found in the physical mechanism of aerosol-cloud interaction and quantifying the related radiative forcing more accurately (Boucher et al., 2013; Dipu et al., 2013). Furthermore, layer detection of tenuous aerosol and
cloud layers and the classification of aerosol subtypes should receive increased attention to improve the accuracy of faint aerosol retrievals. More effective data denoising processes can also be investigated to reduce biases in extinction retrieval, such as the systematic positive bias in the retrieval of daytime observations from CALIPSO.

**Data availability:** The datasets used in this study can be accessed from websites listed in Acknowledgements or by contacting the corresponding author.

**Author contribution:** F.M., Z.P. and D.R. conceived the study. F.M., R.S. Z.P. and D.R. designed the retrieval methodology and comparison experiments. R.S. and Z.P implemented the methodology and carried out the data analysis. LZ, Y.Z. and X.L. contributed to the discussion of methodology and scientific significance. All co-authors commented on and reviewed the manuscript. RS prepared the manuscript with contributions from F.M. and the other co-authors.

**Competing interests:** The authors declare that they have no conflict of interest.

**Acknowledgments:** This study is supported by the National Natural Science Foundation of China (41971285 and 41627804) and the National Key Research and Development Program of China (2017YFC0212600). The numerical calculations in this study have been conducted on the supercomputing system in the Supercomputing Center of Wuhan University. We thank the science teams for providing excellent and accessible data, including CALIPSO (https://asdc.larc.nasa.gov/project/CALIPSO) and SAGE III (https://sage.nasa.gov/missions/about-sage-iii-on-iss/).

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
