# Peer review of "Retrieving Instantaneous Extinction of Aerosol Undetected by CALIPSO Layer Detection Algorithm"

_Atmospheric Chemistry and Physics, 2022_

## Referee Comment (RC3)

The authors present a methodology for converting CALIPSO L1 data to a 20 km averaged backscatter coefficient at 532 nm ($k_{532}$) and compare that product to the L3 CALIPSO extinction coefficient product. The stark difference between the 2 products is the resolution: the L3 product is averaged over a month time period and averaged over 5°x 20°(latitude x longitude). The authors also compare their product to the SAGE III/ISS extinction coefficient at 520 nm ($k_{520}$) for daytime and nighttime CALIPSO observations. Their $k_{532}$ product agreed better with SAGE when the CALIPSO data were collected during the night and performed substantially worse for daytime observations. Overall, per their Fig. 5, it appears that the agreement between the 2 instruments is good.

While I believe there may be value in this methodology, the authors do not present a convincing case. Overall, the paper is poorly written in both organization and especially detail. In its current state it is challenging to understand what the authors did (and why) and the methodological details are insufficient to reproduce the work. Further, the authors do not make any attempt to explain how this methodology is different from previous work (they even cited these previous studies so it should be straight forward for them to explain how their method is different). They simply state that their product is below the 0.01 km$^{-1}$ LOD threshold for the previously-published CALIPSO L2 products. However, I have to question whether this is a fair comparison (i.e., their L1-based product with the L2 product). I think that, as written, the paper does not present a novel approach that will produce a scientifically interesting product and there are some substantial technical issues at play as well. Finally, the authors list 3 main conclusions from this study. However, for reasons listed below, I find all 3 conclusions to be tenuous (or demonstrably wrong), which leave me questioning the scientific merit of this study. For these reasons, and those enumerated below, I cannot recommend this paper for publication at this time.

**1 Major Issues**

The following issues should be addressed prior to resubmission.

1. The paper requires major revisions in writing. Many sentences do not make sense, some references do not make sense within the context of the sentence, and some references do not support what the authors claim.

2. The authors do not make it clear how their method is different from previous work, which leaves me questioning the scientific merit of this work.

3. This work is ultimately a statistical analysis and the authors are looking for faint signal that, per their paper, is below the previously published limit of detection (LOD) for CALIPSO. Their retrieval requires an iterative process but do not attempt to propagate uncertainties through this process. Therefore, we have no was of determining whether their end product is significantly different from zero.

4. One has to wonder how valid a comparison between an instantaneous product and a monthly average is (i.e., the authors L1 product and the standard CALIPSO L3 extinction product)

per their Figure 7. Further, the faint aerosol layer they identify is at ≈17.5 km and they showed the L3 product at 15.2 km. The validity of this comparison is tenuous even when the altitude match, and much less so when the altitudes do not match.

5. It seems the authors are unaware of the 2019 Raikoke eruption, which is likely the source of aerosol in their Figs. 3, 4, 7. Instead, they attribute these layers to a single wildfire event in Siberia without providing any supportive evidence. The papers they cite in support of this claim do not, in fact, support their assertion. Ohneiser et al. 2021 claimed to see smoke up to 13 km from 50°N to 80°N. Kostrykin et al. 2021 did a modeling study wherein they claimed to trace smoke up to ≈ 6 km. The authors claim to see smoke from 10°N to 60°N at 15–20 km. This is in stark contrast to the finding of the 2 referenced works. The authors either need to support their assertion that smoke was present, and responsible for the faint aerosol layer they observed, or reconsider the source.

6. Section 3.1: The authors claim these faint aerosol layers are the product of wildfire events (i.e., smoke). Smoke has a very different lidar ratio from $50~\text{sr}^{-1}$. Did the authors use a lidar ratio of $50~\text{sr}^{-1}$ here? If so, how does this impact the interpretation of the results (e.g., do the faint aerosol layers disappear if the correct lidar ratio is used)?

**2 Minor Issues**

The following issues should be addressed prior to resubmission.

1. The authors perform their coefficient correlation and RMSE calculations in logarithmic space. Why use logarithmic space when this effectively reduces the comparison to essentially how well the orders of magnitude agree.

2. How is "faint aerosol" defined? What qualifies as a "faint aerosol"?

3. On lines 81–83 the sentence containing "...observed by the rays passing through the atmosphere..." does not make sense and this is not how SAGE quantifies extinction coefficient. I think, as written, this sentence is incorrect. Light passes through the atmosphere and is attenuated by some combination of scattering and absorption of molecules, particles, and clouds. The molecular number densities and extinction coefficients are then derived based on the recorded spectra.

4. Throughout the paper the authors refer to SAGE extinction, but do not specify wavelength. They stated within the text that they used the 521 nm channel, but it would help the reader if the wavelength is included in all subsequent references to SAGE extinction.

5. The authors used the SAGE III/ISS v5.1 product while the v5.2 product is available. Vertical smoothing for the extinction products was turned off for the v5.2 product, which effectively provides data at a higher vertical resolution. The authors are encouraged to use this data set since the higher resolution will make their comparisons more meaningful.

6. Line 95 What is the "vertical moving filtering" that was done? What kind of filter is this and what is it removing?

7. Not all of the variables in equations 1-3 are defined within the text (some are defined within a figure, some within a figure caption, some within the text). Please define all variables within the text.

8. Lines 120–121: Please provide reference for $\beta(0) = 0, T_p(0) = 1$.

9. Line 124: What is meant by "same day"? Is this the same calendar date, or within 24 hours?

10. Line 125: You state the SAGE horizontal resolution is ≈300 km; please provide a reference for this value. This raises a greater point: the viewing geometries and sampling volumes of SAGE and CALIPSO are vastly different and may provide another source of uncertainty.

11. Line 126: The reference to Adams et al. 2013 does not make sense here. Please include why this is relevant or remove the reference.

12. Line 127: What is meant by "...where the CALIPSO orbit exceeds 0.75°latitude..."? What is meant by 0.75°? This is confusing, please clarify.

13. Line 144: The authors refer to a "red dashed box" in Fig. 3 (a), but there is no such box in that figure. Please add the box.

14. Lines 148–150: One has to ask, if these faint aerosol layers are visible within the ASR product, then why go to the trouble of calculating extinction coefficient? It seems that computing extinction coefficient introduces unnecessary uncertainties, so my 2 questions are: 1. why use the extinction coefficient?, 2. are the uncertainties that the extinction calculation introduced acceptable?

15. Line 156: It seems the authors are introducing an additional step in their processing algorithm (i.e., the 3x3 mean filtering window). This should be included in the main body of the text and explain what this is.

16. Line 163: Perhaps the pdf did not build properly, but there is no red dash line in Fig. 4 (a) though there is a pink line.

17. Figure 4: What is the vertical pink dashed line?

18. Line 172 and Figure 5 (a): How were these averages calculated? There are 11 points on this line, but the SAGE data exists over a continuum, so there must have been some binning along the x-axis

19. Line 172 and Figure 5 (a): The authors refer to this as "faint aerosol extinction", but does this really qualify as "faint" when the SAGE extinction coefficients exceed 1E-3 (even 1E-2!)?

20. Line 173: The authors claim the CALIPSO retrieval has a low bias while their line of best fit falls just above the 1:1 line, indicating the CALIPSO retrieval has a high bias. Can the authors explain what they mean by low bias and justify this designation?

21. Line 184: What is meant by "mean values of 5% quantiles"? It seems the authors have introduced another calculation that is not defined/explained within the text and is not readily understood by the reader. This should be explained within the text.

22. Line 193: The authors state "...the CALIPSO retrievals in the troposphere should be more reliable [than SAGE]" While I agree about this statement to some degree, this is a blanket statement that is made without qualifier and is misleading. The authors need to explain what they mean here. Further, there are nuances that are important. In the troposphere you can have highly variable lidar ratios; how does this influence the "reliability"? You have to propagate your errors all the way through the stratosphere before hitting the troposphere; how does this influence reliability? Finally, the troposphere is often loaded with aerosol (as compared to the stratosphere), so how often do you expect to be below the 0.01 km$^{-1}$ LOD (i.e., is you method still necessary at this point)? It seems like the discussion regarding the troposphere is misplaced within this paper.

23. Line 201: What is the purpose of the Kim et al 2017 reference here?

24. Section 3.3: The authors attribute the faint aerosol during this time period to the Siberian wildfires. As discussed above this is more likely the impact of the Raikoke eruption. It is strange that this was not mentioned here. Please see discussion above regarding this topic and address the issues stated therein.

25. Figure 7: It will help the reader if panels (a), (b), (c), and (d) all have the same colorbar scale and limits.

26. Line 214: The Kostrykin reference in not germane to this paper as they showed no indication of the Siberian wildfire smoke leaving the troposphere. They estimated a 5% chance of smoke making it to 6 km.

27. Line 224: "...classified as elevated smoke by VFM). This is categorically wrong. The CALIPSO vertical feature mask identifies this plume as "sulfate/other" **not** smoke.

28. Lines 227–228 and Figure 7: It is difficult to compare a monthly average and an "instantaneous" measurement. However, the layer referenced in Fig. 7 (d) is not at the same altitude as what is shown in Fig. 7 (b). Going off Fig. 7 (b) alone, I would expect little (no?) enhancement at 15.2 km for most of the the NH. If we accept Fig. 7 in its current state we cannot draw a decisive conclusion (panels (b) and (d) contradict each other). However, if panels (a, b) were updated to mate the altitude of the layer in (d) this would significantly bolster your case.

29. Line2 242–246, conclusion #1: As stated above, some of your fundamental assumptions will be different within the troposphere. Further, the discussion of tropospheric aerosols does not seem to fit within the context of this paper. Further, the authors failed to demonstrate that this method is novel from previous work. Therefore I find this conclusion to be tenuous.

30. Lines 247–248, conclusion #2: While the authors demonstrated this they were not the first to determine this so I do not view this as a significant finding/conclusion to be drawn from this paper.

31. Lines 249–252, conclusion #3: While the authors demonstrated a faint aerosol layer that extended from 10°N to 60°N (at 20 km) the wrongly classify this as smoke. Further, they claimed the corresponding $k_{532}$ was significantly below the CALIPSO L2 LOD (LOD is 0.01 km$^{-1}$, their aerosol layer was $\approx$0.003 km$^{-1}$) and they failed to account for the propagation of error in their extinction coefficient retrieval. Therefore, we do not know if the layer

they identified is significantly different from background. While I suspect this layer is real, it is difficult to interpret the significance of this conclusion without having this thorough statistical support.

**3 Wording Issues**

The following issues should be addressed prior to resubmission.

1. Line 13: " the susceptibility of clouds to aerosols is more pronounced when the aerosols are faint." This sentence does not make sense. Please clarify meaning.

2. Line 39: What is the "undetected tenuous aerosol layer"? This needs defined.

3. Line 48: The word "tenuous" does not make sense here. Please rephrase.

4. Lines 60–63: This sentence does not make sense. Are the "faint aerosols" composed of both background AND undetected aerosol layers? If so, what does it mean for a particle to be composed of undetected aerosol layers?

5. Lines 83–84: You discuss the extinction coefficient uncertainties of SAGE III/ISS and cite Thomason et al. 2010. The Thomason paper is in reference to SAGE III/METEOR (or SAGE III/M3M). Please indicate that this reference is for SAGE III/M3M and not SAGE III/ISS.

6. Please update all "SAGE III-ISS" to "SAGE III/ISS".

7. I had to read the first paragraph of section 2.3 several times before I understood your methodology. It will help the reader if you change the last sentence of this paragraph to: "To improve the signal-to noise ratio (SNR) of the total attenuated backscatter (TAB) data, we performed the following pre-processing steps prior to running the data through the algorithm presented in Fig. 1:"

8. Line 102: "particle particulate multiple scattering factor" is called "multiple scattering coefficient" at other places in the manuscript. Please make this consistent throughout.

9. Line 124: "diurnal variation of faint aerosols in SAGE III-ISS data" does not make sense. Perhaps you meant "...dirunal variation of faint aerosol within the stratosphere"?

10. Figure 5 caption: Much of this caption is difficult to understand and should be rewritten for clarity. Some information should be removed and put in the text and the authors should organize the caption in such a way as to explain what each panel is in a successive manner. In its current state the caption explains panels (a), (b), (c), then returns to discussion panel (a) without telling the reader what is going on.

11. Line 190: This sentence ("The number of matched points...") does not make sense. Please reword this to better communicate what you did.

12. Lines 194–195: This last sentence does not make sense (what is un-performed?). Please rewrite this to better communicate what you intend.

13. Line 210: Please include the year (e.g., June and August 2019).

14. Lines 229–230: This sentence is difficult to understand. What is a "propagation trajectory" and how can you estimate that based on a single CALIPSO granule? Please rewrite this sentence to better explain what you mean.

15. Line 239: Perhaps the authors meant "...retrieved instantaneous extinction coefficients for faint aerosol layers based on..."?

---

## Author Comment (AC1)

**Manuscript ID:** acp-2022-56

**Original title:** CALIPSO Retrieval of Instantaneous Faint Aerosol

**Revised title:** Retrieving Instantaneous Extinction of Aerosol Undetected by CALIPSO Layer Detection Algorithm

**General comment**

In this manuscript, the authors use the instantaneous observations of CALIOP to retrieve faint aerosols missed by CALIOP's official aerosol layer and profile products. Comparison analyses with SAGE III aerosol product demonstrate good agreement from the middle troposphere to the stratosphere. Also, in the 2019 Siberian fire event, retrieved instantaneous aerosol provided more information on faint aerosol propagation trajectory with higher spatial and temporal resolution than CALIPSO Level 3 monthly-averaged aerosols product.

This study is very interesting because most of the previous studies focused on aerosol and cloud layer retrieval, but the authors propose a novel method to retrieve the generally-ignored faint background aerosol based on CALIOP instantaneous observations. The manuscript is well-written and straightforward. The results are satisfactory and effectively described. This method is expected to provide new data for the investigation of aerosol-cloud interaction, which may offer new insights into the aerosol climate effect that otherwise cannot be seen by studies based on integrated or surface aerosol information (i.e., AOD). Therefore, I suggest this manuscript be published after minor revisions.

*Response*: Thank you for your careful review. We have made great efforts to address your comments, and improved this manuscript largely. The main changes are as follows:

(1)  The title of the paper has been revised as "*Retrieving Instantaneous Extinction of Aerosol Undetected by CALIPSO Layer Detection Algorithm*" to highlight the objectives of the study.

(2)  A new method to retrieve lidar ratios by using SAGE III/ISS products as a constraint has been added to the manuscript (Section 2.4). We add the comparison of retrieved undetected aerosol extinction based on globally SAGE-constrained and fixed lidar ratios in Section 3.4 to highlight the effect of lidar ratio.

(3)  Uncertainties in the extinction coefficient retrieval were calculated to assess the reliability of the extinction results of undetected aerosol.

(4)  The Raikoke eruption event is added to the comparison of instantaneously retrieved undetected aerosol extinction and Level 3 product in Section 3.3.

**Major comments**

1. The introduction needs to become more refined and better linked to the scientific literature. It should be clearer where the gaps are in the literature and what the contribution of this study is in this respect.

   *Response*: The introduction has been revised to highlight the research objectives and significance of this paper. The main changes are as follows.

   (1) The statements in this section have been polished to make the background clear.

   Few studies focus on retrieving aerosols undetected by the CALIPSO detection algorithm (Kar et al., 2019; Kim et al., 2017). Kim et al. (2017) attempted to calculate the missing AOD of these undetected aerosols by constraining of the MODIS AOD over ocean. However, that study mainly focused on the AOD of the undetected aerosol with a fixed lidar ratio, but the extinction of the undetected aerosol was rarely discussed and verified. Additionally, Kim et al. (2017) provided the same lidar ratio (28.75 sr) for the troposphere and stratosphere globally, potentially introducing large uncertainty for extinction retrieval. Recently, the CALIPSO Level 3 Stratospheric Aerosol Profile product was released. However, the purpose of CALIPSO Level 3 products is to provide monthly grid data (5°×20° in latitude and longitude) (Kar et al., 2019), which are insufficient to support studies sensitive to temporal and spatial variations of aerosols, such as studies of aerosol and cloud interactions (Ma et al., 2015). Furthermore, many studies suggest that CALIPSO may potentially obtain more information on faint aerosols with appropriate data processing (Thomason et al., 2007; Vernier et al., 2009; Kar et al., 2019).

   (2) The summary of the research has been rewritten to highlight the innovation and significance of the study, as follows:

   Thus, the present study focuses on retrieving the instantaneous extinction of aerosol undetected by the CALIPSO detection algorithm based on the single-track CALIPSO data. The global distribution of the lidar ratio is obtained with the constraint of the Stratospheric Aerosol and Gas Experiment III on the International Space Station (SAGE III/ISS) observation in the troposphere and stratosphere, respectively. Furthermore, the CALIPSO nighttime and daytime extinction coefficients are retrieved and compared against independent SAGE III/ISS data and CALIPSO Levels 2 and 3 aerosol products. Finally, the impacts of the retrieved lidar ratio and empirical lidar ratio are discussed.

2. The authors use a lidar ratio of 28.75 sr in the troposphere, referring to Kim et al. The lidar ratio is one of the key parameters for aerosol extinction retrieval and varies with aerosol type. Therefore, I suggested the authors more deeply discuss and analyze the lidar ratio, including the difference between land and ocean.

***Response***: Thanks for your suggestion. In this study, a new method is utilized for lidar ratio retrieval by using SAGE III/ISS AOD as a constraint globally (including over land and ocean). The comparison shows good agreement with the independent SAGE III/ISS aerosol extinction (R=0.66). Especially, the relative uncertainties of the retrieved extinction coefficients at $10^{-3}$ km$^{-1}$ and $10^{-4}$ km$^{-1}$ are 35% and 125%, respectively, while the minimum extinction of CALIPSO L2 is 0.01 km$^{-1}$ with 40% uncertainty (Kacenelenbogen et al., 2011; Toth et al., 2018; Winker et al., 2013; Winker et al., 2009). The new method is described in Section 2.3, as follows.

[revised manuscript text omitted]

3.  The CALIPSO retrieval of instantaneous faint aerosols is very challenging. Is there a useful way to improve the retrieval in the future, such as using a wavelet to denoise the CALIPSO level 1 data before retrieval? Although I do not recommend using denoising algorithms in the study of this paper because we prefer to do original research using the most formal methods first, I suggest discussing it for guiding future work.

***Response***: Thanks for your suggestion. Since the SNR of undetected aerosols in CALIOP instantaneous observations is very low, the denoising process is necessary to enable accurate retrieval, we have used a data pre-processing method consistent with Kim et al. (2017). As you say, a more efficient and disciplined denoising process is needed in the future and we have added a discussion of this in the article, as follows:

More effective data denoising processes can also be investigated to reduce biases in extinction retrieval, such as the systematic positive bias in the retrieval of daytime observations from CALIOP.

**Specific comments**

1. Line 23: "capture" should be "captures".

   ***Response***: Modified as per your suggestion.

2. Line 38: it is not always true to argue that the aerosol particules in the PBL "can usually be detected by CALIPSO", which is inconsistent with previous findings. Therefore, this statement can be rephrased as "can only be detected by CALIPSO in the upper PBL in the absence of cloud (doi: 1016/j.atmosres.2016.05.010)"

   ***Response***: Modified as per your suggestion, as follows:

   Aerosols are mostly concentrated in the planetary boundary layer (PBL), where optically thick aerosol layers occur and can usually be detected by the CALIPSO detection algorithm (Li et al., 2017; Kim et al., 2021; Guo et al., 2016).

3. Line 43: The citation may be corrected.

   ***Response***: This paragraph has been reorganized and this sentence has been deleted.

4. Line 49: Clouds interact directly with surrounding aerosols, and in particular sub-cloud aerosols have a more significant effect on cloud production. However, these aerosols are not exactly the same as what the authors refer to as faint aerosol. A more rigorous and accurate expression is recommended.

   ***Response***: Revisions have been made, as follows:

   However, clouds interact directly with ambient sub-cloud aerosol instead of near-surface heavy aerosol, the properties of which could be very different, especially for aerosol and ice cloud interactions (Rosenfeld et al., 2014).

5. Line 50: This study is not motivated by the aerosol proxy used for aerosol-cloud interaction. I think the ignorance of faint aerosol surrounding high-altitude cloud layers is the culprit to complex the quantification of aerosol climate effect. Therefore, "an improper aerosol proxy (such as AOD)" can be changed to "the ignorance of faint aerosols surrounding high-altitude cloud layers" or something like this.

   ***Response***: Thank you for your suggestion, and the word has been revised as follows:

   Thus, the ignorance of faint aerosols surrounding high-altitude cloud layers causes large uncertainty in quantifying the climate effect of aerosols.

6. Line 68: "in" should be "since".

   *Response*: Modified, as follows:

   The CALIPSO mission introduced new technology for retrieving aerosol profiles from space since April 2006,…

7. Line 73: It is recommended use the full name of the product (e.g. VFM) where it first appears in the manuscript.

   *Response*: Thank you for your suggestion, we have used the full name where it first appears in the revised manuscript, as follows:

   These aerosol layers should belong to a continuous one (shown in the red dashed box), but the Vertical Feature Mask (VFM) does not show the aerosol ($\sim$0.01 km$^{-1}$) between the two strong aerosol layers ($\sim$0.03 km$^{-1}$),…

8. Line 97: "vertical" should be "vertically".

   *Response*: This is no longer an issue because the corresponding sentence has been changed to:

   Referring to Kim et al. (2017), the TAB profiles are reduced to a vertical resolution of 300 m by linear interpolation to improve the SNR, …

9. Line 105: How is the SNR calculated here based on this formula?

   *Response*: In the revised manuscript it is further explained how the SNR is calculated and the statements are moved to Section 3.2 containing the SNR distribution graph, as follows:

   The SNR is calculated according to $SNR = \mu \,/\, \sigma$ based on the pre-processed TAB with 20 km horizontal resolution in the matching range, where $\mu$ and $\sigma$ are the mean and the standard deviation of the signal, respectively.

10. Line 126: The matching of CALIPSO and SAGE considers spatial distances. What about temporal distances?

    *Response*: CALIPSO and SAGE observations within the same dates are matched, as specified in the revised manuscript as follows:

    Since only daytime data from SAGE III/ISS are available, the CALIPSO orbits are spatially and temporally matched to the nearest SAGE III/ISS observations on the same calendar date with the consideration for a smaller temporal-spatial variation of faint aerosol comparing strong aerosol at the near-surface.

11. Line 144: The "red dash boxed area" is not marked in the figure 3a.

    ***Response***: Modified, as follows:

[Figure]

Figure 7. (a) MODIS Terra true-color image in the daytime and the passing CALIPSO track (yellow line) at night. (b) Latitude–altitude aerosol extinction of the corresponding nighttime CALIPSO track, same as in Figure 4a. The purple and black boundary lines and extinction inside represent the detected aerosol and cloud layers provided by CALIPSO Level 2 products, respectively. (c) Aerosol subtypes in CALIPSO VFM product (N/A=not applicable, 1=marine, 2=dust, 3=polluted continental/smoke, 4=clean continental, 5=polluted dust, 6=elevated smoke, 7=dusty marine, 8=PSC aerosol, 9=volcanic ash, 10=sulfate/other). (d) Attenuated scattering ratio.

12. Line 163: Suggest reword "shows well the consistency" to "shows high consistency".

    ***Response***: Modified, as follows:

    Figure 4b shows high-consistent extinctions of CALIPSO undetected aerosol and matched SAGE III/ISS 521 nm aerosol product (red dash line in Figure 4a) above 15 km.

13. Line 226: The range described here (60-10°N) does not correspond to the range of the red rectangular box in the diagram (40-10°N), and it is suggested to keep it consistent.

    ***Response***: Thanks for your careful review, it has been revised as follows:

    As shown in the red dash box of Figure 8d, aerosol extinction enhancement (~0.005

km$^{-1}$) occurs around 17 km near 40°N to 5°N,…

14. Line 240: "and compared them" can be changed to "which are compared"

    ***Response***: Thanks for your suggestion but this paragraph has been reorganized and this sentence has been deleted.

15. English should be further improved by a native English speaker.

    ***Response***: Thanks for your suggestions, the English in the revised manuscript has been improved.

**References**

[revised manuscript text omitted]

---

## Author Comment (AC2)

**Manuscript ID:** acp-2022-56

**Original title:** CALIPSO Retrieval of Instantaneous Faint Aerosol

**Revised title:** Retrieving Instantaneous Extinction of Aerosol Undetected by CALIPSO Layer Detection Algorithm

**General comment**

The space-borne lidar CALIOP on board CALIPSO satellite has been providing global measurements of aerosol and cloud backscatter profiles since 2006. Successive improvements in algorithms and calibration procedures have resulted in several versions of the products. Nonetheless there is always scope for further improvements. In particular, it has been known that aerosols with weak backscatter sometimes fall below the CALIOP layer detection threshold and several works have focused on this aspect as has been pointed out by the authors.

In this paper the authors attempt to retrieve what they call "instantaneous faint" aerosols using the CALIOP backscatter data in both the stratosphere and troposphere and compare the results with SAGE III on ISS. Firstly it is not clear to me what the authors mean by "faint" aerosols---are these background aerosols or aerosols that just fall below the detection threshold but otherwise retain the intensive optical properties of the nearby detected layers. The methodology is nothing new and Kim et al. (2017) have used the same 20 km horizontal and 300 m vertical averaging for their extinction retrievals. The Kim et al. (2017) paper did a much more comprehensive study of the weak aerosols unlike this manuscript which lacks in some important details and other aspects, as described below. I do not believe this paper presents enough innovative ideas or interesting new results that will be useful to the community. I regret that I am not able to recommend this manuscript for publication in Atmospheric Chemistry and Physics. I have the following comments in no particular order:

***Response***: Thank you for your careful review. We have made great efforts to address the issues you raised and highlight our significance, and the manuscript has been largely improved. The main changes are as follows.

(1) The title of the paper has been revised as "*Retrieving Instantaneous Extinction of Aerosol Undetected by CALIPSO Layer Detection Algorithm*" to highlight the objectives of the study.

(2) A new method to retrieve lidar ratios by using SAGE III/ISS products as a constraint has been added to the manuscript (Section 2.4). This study no longer uses the fixed lidar ratio (i.e. 28.75 sr) as used in Kim et al. (2017). We add the comparison of retrieved undetected aerosol extinction based on globally SAGE-constrained and fixed lidar ratios in Section 3.4 to highlight the effect of the lidar

ratio.

(3) Uncertainties in the extinction coefficient retrieval were calculated to assess the reliability of the extinction results of undetected aerosol.

(4) The Raikoke eruption event is added to the comparison of retrieved undetected aerosol extinction and Level 3 product in Section 3.3.

(5) The results, discussion and conclusions were largely rewritten in the revised manuscript to highlight the significance of this study.

**Major comments**

1.  The authors seem to use a constant altitude bin of 300 m to average the level 1 (L1) backscatter profiles in the entire altitude range from 0-36 km. This is not tenable, because CALIPSO L1 backscatter profiles have varying resolution with altitude, with 30 m from surface to 8.3 km, 60 m between 8.3-20.2 km, 180 m between 20.2-30.1 km and 300 m above 30.1 km. Properly accounting for these differences will require setting the binning at 900 m as was done in Kar et al. (2019) for the level 3 stratosphere aerosol product.

    *Response*: Sorry for the confusion. In the revised manuscript we have added a description of the vertical resolution of the CALIPSO Level 1 data and explained that we reducing the vertical resolution to 300 m by linear interpolation in Section 2.1.

    (3) The vertical resolution of the CALIPSO Level 1B TAB profiles varies with the height of 30, 60, 120, and 300 m for −0.5–8, 8–20.2, 20.2–30.1, and 30.1–40 km, respectively. Referring to Kim et al. (2017), the TAB profiles are reduced to a vertical resolution of 300 m by linear interpolation to improve the SNR, followed by a vertical moving mean filtering (with a 5-point window) and horizontal averaging to 20 km to retrieve the extinction of undetected aerosol.

2.  I am surprised that the authors are talking about the "instantaneous faint" aerosols in the extinction range $10^{-4}$-$10^{-5}$ km$^{-1}$ and yet do not mention the estimated uncertainties for these extinction profiles at all. When they are claiming to do better than the standard CALIPSO level 2 (L2) and level 3 (L3) products, they should discuss the resulting uncertainties for their extinctions to bolster their claims. Kim et al. (2017) had discussed in detail the uncertainties in their retrievals particularly those coming from the lidar ratios used. The lack of any such discussion in this paper is a major drawback.

    *Response*: We have added the calculation of the extinction uncertainty for the retrieval in Section 2.4, as follows:

For the retrieved extinction of undetected aerosol, we calculated the uncertainty to assess the reliability of the results according to the algorithm of CALIPSO Level 2 aerosol product (Young et al., 2013), where the main equations are as follows:

$$\frac{\Delta \beta'_N(r)}{\beta'_N(r)} = \left\{ \left[ \frac{\Delta \beta'(0,r)}{\beta'(0,r)} \right]^2 + \left[ \frac{\Delta C_N(r_N)}{C_N(r_N)} \right]^2 \right\}^{1/2}, \tag{5}$$

$$\left( \Delta \beta_p(r) \right)^2 = \beta_T^2(r) \left[ \left( \frac{\Delta \beta'_N(r)}{\beta'_N(r)} \right)^2 + \left( \frac{\Delta T_M^2(r_N,r)}{T_M^2(r_N,r)} \right)^2 + \left( \frac{\Delta T_P^2(r_N,r)}{T_P^2(r_N,r)} \right)^2 \right] + \left( \Delta \beta_M(r) \right)^2, \tag{6}$$

$$\Delta \sigma_P(r) = \left[ \left( \frac{\Delta S_p}{S_p} \right)^2 + \left( \frac{\Delta \beta_p(r)}{\beta_p(r)} \right)^2 \right]^{1/2} \sigma_P(r), \tag{7}$$

where $\Delta \beta_p(r)$ and $\Delta \sigma_P(r)$ in Eqs (6) and (7) are the particle backscatter uncertainty and particle extinction uncertainty, respectively; they are the target parameters for the calculation. Eq. (5) is the formula for one of the terms of Eq. (6), where $\Delta \beta'_N(r)$ is the uncertainty of the renormalized TAB, $\Delta \beta'(0,r)$ is the uncertainty of the TAB, and $\Delta C_N(r_N)$ is the uncertainty of renormalization. The error due to renormalization is negligible (Kim et al., 2017) because the starting altitude of retrieval ($r_N$ =36 km) is consistent with the calibration region (36–39 km) for the CALIPSO Level 1B Version 4 product (Kar et al., 2018); therefore, $\Delta C_N(r_N)$ is set to 0. The standard deviation of the TAB is used to approximate $\Delta \beta'(0,r)$ because the TAB in this study was pre-processed.

Uncertainty is found in the calibration factor in $\Delta \beta'(0,r)$, which contains systematic and random components (Young et al., 2013), and this approximation neglects the systematic error in the calibration factor, producing a low bias in the uncertainty calculation. Fortunately, the calibration factor bias of the nighttime CALIPSO V4 product has been reduced to $1.6\% \pm 2.4\%$ (Kar et al., 2018). Additionally, Kim et al. (2017) pointed out that the bias caused by the lidar ratio is dominated in the retrieval. Thus, we consider ignoring the calibration factor in the systematic error. The other terms in Eq. (6), total backscatter coefficient ($\beta_T(r)$), molecular and particle two-way transmittance uncertainty ($\Delta T_M^2(r_N,r)$ and $\Delta T_P^2(r_N,r)$), and molecular backscatter uncertainty ($\Delta \beta_M(r)$) are calculated in the same way as in Young et al. (2013) and are not repeated here. $S_p$ and $\Delta S_p$ in Eq. (7) are selected from the median and median absolute deviation, respectively, in the retrieved 20°×20° grid lidar ratio based on CALIPSO profile locations.

Also, we discuss the uncertainties corresponding to the different magnitudes of extinction in the retrieval in Section 3.2, as follows:

The averaged black line in Figure 5b show the mean relative uncertainties of CALIPSO, specifically ~35% and ~125% for the retrieved extinction of $10^{-3}$ and

10-4 km-1, respectively.

[Figure]

Figure 5. (a) Correlation plots of the retrieval within the matching grid of CALIPSO nighttime and SAGE III/ISS product from 5 km to 30 km for 12 months of validation. The color bar represents the sample size. The black bins represent the mean values of each 10% quantile (0-10%, 10-20%...and 90-100%) of SAGE III/ISS 521 nm aerosol product and corresponding CALIPSO retrieval. The I-type bars indicate the standard error of each 10% quantile CALIPSO retrieval. (b) The relative uncertainty of one-degree CALIPSO extinction.

3.  In their retrieval algorithm they mention excluding the aerosol and cloud layers detected by the CALIOP L2 algorithm and all data below those layers similar to the methodology employed in Kar et al. (2019). However one important filter has been left out which has to do with the thin cirrus clouds. These clouds often fall below the layer detection threshold and can significantly contaminate the "faint" aerosol profiles they are trying to retrieve particularly in the UTLS area. Kar et al. (2019) had used a filter on the volume depolarization ratio to take out these ice clouds. In fact cloud clearance can be an issue with SAGE occultation measurements as well, which affect the data below 20 km (Thomason and Vernier, 2013, https://doi.org/10.5194/acp-13-4605-2013). It is not clear if the authors have used any filter for cloud-clearing of SAGE III data. Further note that the SAGE III-ISS 521 nm extinction product has low bias between 20 km and 25 km at mid latitudes possibly relating to the ozone interference (Knepp et al., 2022, https://doi.org/10.5194/amt-2021-333). These issues should have repercussions for the CALIPSO/SAGE III comparisons the authors have attempted.

    ***Response***: Good suggestions! We have added the removal of thin cirrus clouds of CALIPSO data in Section 2.1, as follows:

    (2)  The clouds and aerosol layer detected by the SIBYL and the data below them

were removed. We used a threshold value of 0.5 in the attenuated color ratio (the ratio of the TAB at 1064 and 532 nm) to remove undetected tenuous cirrus clouds, similar to the data screening method of the CALIPSO Level 3 Stratospheric Aerosol Profile product (Kar et al., 2019).

The cloud-contaminated data in SAGE III/ISS was removed in Section 2.2, as follows:

We removed the bins in the SAGE III/ISS aerosol extinction profile with color ratio (the ratio of the aerosol extinction at 521 and 1022 nm) in the range of 0.8 to 1.2 to avoid cloud contamination (Schoeberl et al., 2021).

The SAGE III/ISS 521 nm aerosol extinction product was corrected with reference to Knepp et al. (2021).

A low bias in the extinction coefficients of the SAGE III/ISS aerosol product is observed at 521 nm due to the ozone interference in the retrieval algorithm. This finding is more pronounced at mid-latitudes and altitudes between 20 and 25 km (Knepp et al., 2021). The following equation is therefore used to correct the extinction at 521 nm (Knepp et al., 2021):

$$log\ \sigma_{521} = \frac{log\ (\frac{\sigma_{450}}{\sigma_{755}}) \times log\ (\frac{521}{755})}{log\ (\frac{450}{755})} + log\ (\sigma_{755}), \tag{1}$$

where $\sigma$ is the extinction coefficient, and the numbers represent the wavelength.

4. The authors use a lidar ratio of 50 sr in the stratosphere and 28.75 sr in the troposphere. In Figure 3, they are retrieving "faint" aerosol in between and in continuity with the smoke layers retrieved by CALIOP L2 product, assuming that the "faint" aerosol is also smoke and referring to "the continuous nature of this aerosol layer". For smoke, CALIOP L2 uses a lidar ratio of 70 sr, so the "faint" aerosol extinctions retrieved using a lidar ratio of 28.75 sr will have a significantly low bias. I am also intrigued by the distinct band of "faint" aerosols above 10 km extending from 15°S-55°S—is this background aerosol, smoke? Indeed how confident are the authors that it is aerosol at all? Similarly in the ASR plot Figure 3e, what are all those clumps between 10-30 km? It all just looks like noise to me although the ASR values are quite significant (~1.3) and about the same as within the pink box. Similarly in Figure 7, the authors are showing the plume of "faint" aerosols "connected with the VFM aerosol features"—i.e Siberian smoke extending from 10oN-60oN. Once again, they should use the more appropriate lidar ratio for smoke (70 sr) rather than that (50 sr) for stratospheric background aerosols. Since the authors use coincident extinction data from SAGE III-ISS, have they tried to obtain the lidar ratio (in Figure 4b, for instance) using the extinction

from SAGE III and backscatter from CALIOP following the method given in Kar et al. (2019)? It will be interesting to see how those compare with the values they are using.

***Response***: You are right, the lidar ratio selection is one of the keys in retrieval. We have improved the lidar ratio selection 
[revised manuscript text omitted]

In addition, we have added the calculation of the extinction uncertainty for the retrieval to indicate how confidently the retrieval is true for the undetected aerosol in Section 2.4. Also, we discuss the uncertainties corresponding to the different magnitudes of extinction in the retrieval in Section 3.2, as described in Comment 2.

5. It seems to me that in Figure 4, the layer at ~15 km is detected and retrieved by standard CALIOP L2---wonder how that compares with the profiles the authors retrieve. In any case the "faint' layers between 15-20 km in this Figure do not look quite faint to me, and is likely the aerosol that just missed the layering threshold of standard L2.

*Response*: Sorry for the confusion, in this study, faint aerosol refers to the aerosol undetected by CALIPSO Layer Detection Algorithm. We now clarify the definition of weak aerosols and give it directly in the revised title.

*Retrieving Instantaneous Extinction of Aerosol Undetected by CALIPSO Layer Detection Algorithm*

6. As the authors have mentioned, CALIOP has been experiencing low energy laser shots since late 2016 primarily impacting the SAA region and accordingly they have excluded the SAA region. However those low energy shots have been spreading to other latitudes as well and can lead to artifacts in the data including false layer detections at all altitudes, particularly in the dayside. These effects can impact the extinction retrievals the authors are attempting and can be alleviated using the prescription given in the data advisory.

*Response*: Thank you for your suggestion, we have removed the affected CALIPSO data according to the data advisory, as follows:

(1) We removed the affected CALIPSO observations according to Low Laser Energy Technical Advisory due to the effects of an elevated frequency of low-energy laser shots of CALIPSO within the South Atlantic Anomaly (SAA) (https://www-calipso.larc.nasa.gov/resources/calipso_users_guide/advisory).

7. Line 173-174—"Further, we can see that the retrieved aerosol extinction is much less than the detection limit (0.01 km-1) of the CALIPSO Level 2 product" and lines 234-235—"Instantaneous retrieval of faint aerosol at 20 km horizontal resolution provides a chance to deeper understand and quantify the aerosol impact on climate beyond the current CALIPSO Level 3 Stratospheric Aerosol Profile product". I think the authors are missing the rationale behind the L2 and L3

products. In my understanding CALIOP L2 first detects a "layer" using a range-dependent threshold and then assigns an aerosol subtype to it, for which a lidar ratio is available. This lidar ratio varies from 23-70 sr. The layer detection scheme for relatively low SNR measurements like CALIOP is quite complicated (Vaughan et al., 2009, doi:10.1175/2009JTECHA1228.1) and was designed to minimize false positive detections which leads to some undesired missed detections. However, overall it has worked very well and without this and (hopefully) proper assignment of lidar ratios for those different types of aerosols, CALIOP products would not be as useful as they have been. The next generation detection scheme for lidars uses 2D algorithms using both 532 nm and 1064 nm backscatter measurements and will lead to much more accurate detections including those of weakly scattering particulates (see Vaillant de Guelis et al., 2021, https://doi.org/10.5194/amt-14-1593-2021). On the other hand, in the L3 stratospheric aerosol product, which is built from the L1 profiles, the retrieved extinctions are of the same order as the authors retrieve here. The L3 product has low spatial resolution ($5^{\circ} \times 20^{\circ}$ in lat/lon necessary to increase the SNR and produces a reliable picture of the known stratospheric features) and is geared towards modelling applications. In other words, the L2 and L3 products have different goals and limitations. L2 products in the stratosphere employ different lidar ratios for different subtypes (ash, sulfates, smoke) unlike in the L3 product where a constant value is used for background as well as the full-aerosol mode. If the authors propose to retrieve the "instantaneous faint" aerosols in between the layers of different subtypes (as in Figures 4 and 7) then they should use the appropriate lidar ratios as mentioned above---this entails using the subtypes defined in CALIOP L2 or they can define their own subtypes.

***Response***: Thank you very much for your suggestions. In the revised manuscript, we have added a description of the scientific process for each level of CALIPSO product to help the reader understand the relationship between our study and the current CALIPSO products (i.e., fill the gap of the high spatial and temporal distribution of undetected aerosols in CALIPSO Level 2 and Level 3 products) in Section 2.1, as follows.

The CALIPSO team has released different levels of products for different scientific objectives. Level 1 products are calibrated observations containing environmental parameters. Level 2 products are physical, chemical, and optical parameters of aerosol layers and cloud layers obtained according to a series of technical routes. The aerosol and cloud layers are firstly detected by the Selective Iterative Boundary Locator (SIBYL) (Vaughan et al., 2009), then classified by the Scene Classification Algorithms (SCA) (Kim et al., 2018), and finally the extinction coefficient is retrieved according to the Hybrid Extinction Retrieval Algorithm (Winker et al.,

2010; Young et al., 2018). Level 3 products provide a monthly averaged gridded global distribution data of clouds and aerosols (Kar et al., 2019).

At the same time, we reorganize the statements you point out here to maintain a correct understanding of the scientific objectives of the CALIPSO product while comparing retrieved results and them in Section 3.2.

The averaged black line in Figure 5b show the mean relative uncertainties of CALIPSO, specifically ~35% and ~125% for the retrieved extinction of $10^{-3}$ and $10^{-4}$ km$^{-1}$, respectively. This indicates the retrieved extinction of undetected aerosol is much smaller than the low boundary of the detected aerosol extinction ($10^{-2}$ km$^{-1}$) from the CALIPSO Level 2 extinction product with a 40% uncertainty (Kacenelenbogen et al., 2011; Toth et al., 2018; Winker et al., 2013; Winker et al., 2009). Similarly, Watson‐Parris et al. (2018) noted through the model that the minimum value of aerosol extinction at 0–15 km should be close to $10^{-4}$ km$^{-1}$, whereas CALIPSO Level 2 aerosol products remain above $10^{-2}$ km$^{-1}$, and the mean fraction of aerosol undetected by CALIPSO daytime (nighttime) retrievals is 92% (87%) globally.

[Figure]

Figure 5. (a) Correlation plots of the retrieval within the matching grid of CALIPSO nighttime and SAGE III/ISS product from 5 km to 30 km for 12 months of validation. The color bar represents the sample size. The black bins represent the mean values of each 10% quantile (0-10%, 10-20%...and 90-100%) of SAGE III/ISS 521 nm aerosol product and corresponding CALIPSO retrieval. The I-type bars indicate the standard error of each 10% quantile CALIPSO retrieval. (b) The relative uncertainty of one-degree CALIPSO extinction.

Furthermore, we discussed the limitations of the use of lidar ratios in the current study, as responded to in Comment 4. We are also working on the layer detection and classification algorithm (Mao et al., 2021), which is very potential to improve

both the retrieval of the detected and the undetected aerosols in the future.

8. Section 3.3. The authors assume all of the stratospheric perturbation in the northern mid/high latitude is coming from the Siberian wildfires. Much of this may actually be from the Raikoke volcanic eruption (June 2019) instead (Kloss et al., 2021, https://doi.org/10.5194/acp-21-535-2021, Knepp et al., 2022, etc.).

*Response*: Thank you for your suggestion. The Raikoke volcanic eruption has been added to the interpretation of stratospheric aerosol enhancement in the Northern Hemisphere in August 2019, as follows:

A significant amount of aerosol enhancement was observed in the stratosphere in August in the northern hemisphere (Figure 8b), possibly due to the eruption of the Raikoke Volcano in June 2019 (Knepp et al., 2021; Kloss et al., 2021).

9. Lines 226-229 and lines 250-252. I don't understand how from one browse image the authors can show "faint" aerosol "propagating" from 60oN to 10oN. By the way the CALIPSO transect shown in Figure 7b passes through the well-known Asian Tropopause Aerosol Layer or ATAL (Vernier et al., 2011, https://doi.org/10.1029/2010GL046614, Fairlie et al., 2020, https://doi.org/10.1029/2019JD031506, etc.) region. How do they know it's all smoke from Siberia (or, sulfates from Raikoke) rather than at least partly being contributed by ATAL? In fact the ATAL feature mostly falls below the CALIOP layer detection and is seen in the adequately averaged L1 data as in CALIOP L3 product.

*Response*: Sorry for the confusion. The figure and related sentence have been revised. And the Raikoke volcanic eruption has been added to the interpretation of stratospheric aerosol enhancement in the Northern Hemisphere in August 2019.

Figures 8a and 8b show the spatial distribution of aerosol extinction averaged in June and August 2019 at 17 km altitude from CALIPSO Level 3 monthly-averaged Stratospheric Aerosol Profile product with the resolution of 5°×20° in latitude and longitude (Kar et al., 2019). A significant amount of aerosol enhancement was observed in the stratosphere in August in the northern hemisphere (Figure 8b), possibly due to the eruption of the Raikoke Volcano in June 2019 (Knepp et al., 2021; Kloss et al., 2021). We selected two CALIPSO tracks across aerosol enhancement areas in June and August (Figures 8c and 8d), respectively. The stratosphere at the northern hemisphere latitudes is clean, whereas natural dust aerosol prevails in the lower troposphere on June 10 when Raikko has not yet erupted (Figures 8c and 8e). The clean condition shown by our retrieval is consistent with the CALIPSO Level 3 products that indicate the clean stratosphere

at a monthly temporal scale.

Following the onset of volcanic eruptions, strong stratospheric aerosol layers are found in the stratosphere between 50°N and 60°N that are classified as sulfate by the VFM (Figure 8f). As shown in the red dash box of Figure 8d, aerosol extinction enhancement (~0.005 km$^{-1}$) occurs around 17 km near 40°N to 5°N, which corresponds to the monthly average scale aerosol contamination in the stratosphere throughout the Northern Hemisphere in Figure 8b, but is not captured by CALIPSO Level 2 products (Figure 8f). Therefore, the retrieved undetected aerosol extinction can well capture the aerosol enhancement from special events at a horizontal resolution of 20 km (Figure 8d). The color ratios, particle depolarization ratios, and integrated attenuated backscatter are extracted manually for the red dashed region (16 km to 20 km, 40°N to 5°N) with an average of 0.17, 0.02 and 0.00033 sr$^{-1}$, respectively. Using these optical and non-optical properties (i.e., center height, temperature and latitude), aerosol subtypes can be determined by the CALIPSO Scene Classification Algorithms (Kim et al., 2018). The results show that the aerosol subtype in this region is sulfate, which supports that the aerosol enhancement is more likely to be from the eruption of the Raikoke Volcano.

[Figure]

Figure 8. (a) and (b) are the stratospheric extinction distributions of CALIPSO Level 3 Stratospheric Aerosol Profile products at 17 km in June and August,

respectively. (c) and (d) are the retrieved aerosol extinction scenes based on CALIPSO instantaneous data on June 10 and August 10, respectively, consistent with Figure 4a. The corresponding trajectories for the two scenes are shown as red lines in (a) and (b), and the corresponding aerosol subtypes are shown in (e) and (f), the same as in Figure 7c.

Additionally, we did not see stratospheric aerosol enhancement in the same regions in 2018 consistent with 2019 (Figure R1) Therefore, we do not think the stratospheric aerosol enhancement in 2019 (Figure 8) is the result of the well-known Asian Tropopause Aerosol Layer.

[Figure]

Figure R1 (a) and (b) are the stratospheric extinction distributions of CALIPSO Level 3 Stratospheric Aerosol Profile products at 17 km in August 2018 and August 2019, respectively.

10. Line 84, line 188—The aerosol product discussed in Thomason et al. (2010) paper related to the SAGE III instrument on Meteor 3M spacecraft, not the ISS.

    ***Response***: Thanks for your suggestion. This paragraph has been reorganized and this sentence has been deleted.

**References**

[revised manuscript text omitted]

---

## Author Comment (AC3)

**Manuscript ID:** acp-2022-56

**Original title:** CALIPSO Retrieval of Instantaneous Faint Aerosol

**Revised title:** Retrieving Instantaneous Extinction of Aerosol Undetected by CALIPSO Layer Detection Algorithm

**General comment**

The authors present a methodology for converting CALIPSO L1 data to a 20 km averaged backscatter coefficient at 532 nm (k532) and compare that product to the L3 CALIPSO extinction coefficient product. The stark difference between the 2 products is the resolution: the L3 product is averaged over a month time period and averaged over $5° \times 20°$(latitude x longitude). The authors also compare their product to the SAGE III/ISS extinction coefficient at 520 nm (k520) for daytime and nighttime CALIPSO observations. Their k532 product agreed better with SAGE when the CALIPSO data were collected during the night and performed substantially worse for daytime observations. Overall, per their Fig. 5, it appears that the agreement between the 2 instruments is good.

While I believe there may be value in this methodology, the authors do not present a convincing case. Overall, the paper is poorly written in both organization and especially detail. In its current state it is challenging to understand what the authors did (and why) and the methodological details are insufficient to reproduce the work. Further, the authors do not make any attempt to explain how this methodology is different from previous work (they even cited these previous studies so it should be straight forward for them to explain how their method is different). They simply state that their product is below the 0.01 $km^{-1}$ LOD threshold for the previously-published CALIPSO L2 products. However, I have to question whether this is a fair comparison (i.e., their L1-based product with the L2 product). I think that, as written, the paper does not present a novel approach that will produce a scientifically interesting product and there are some substantial technical issues at play as well. Finally, the authors list 3 main conclusions from this study. However, for reasons listed below, I find all 3 conclusions to be tenuous (or demonstrably wrong), which leave me questioning the scientific merit of this study. For these reasons, and those enumerated below, I cannot recommend this paper for publication at this time.

*Response*: Thank you for your careful review. We have made great efforts to address the issues you raised, and the manuscript has been largely improved. The main changes are as follows.

(1) The title of the paper has been revised as "*Retrieving Instantaneous Extinction of Aerosol Undetected by CALIPSO Layer Detection Algorithm*" to highlight the

objectives of the study.

(2) A new method to retrieve lidar ratios by using SAGE III/ISS products as a constraint has been added to the manuscript (see more details in Section 2.4). This study no longer uses the fixed lidar ratio (i.e. 28.75 sr) as in previous study of Kim et al. (2017). We also add a comparison of retrieved undetected aerosol extinction based on globally SAGE-constrained and fixed lidar ratios in Section 3.4 to highlight the effect of the lidar ratio we updated.

(3) Uncertainties in the extinction coefficient retrieval were calculated to assess the reliability of the extinction results of undetected aerosol. The comparison shows good agreement with the independent SAGE III/ISS aerosol extinction (R=0.66). Especially, the relative uncertainties of the retrieved extinction coefficients at $10^{-3}$ $km^{-1}$ and $10^{-4}$ $km^{-1}$ are 35% and 125%, respectively, while the minimum extinction of CALIPSO L2 product is 0.01 $km^{-1}$ with 40% uncertainty (Kacenelenbogen et al., 2011; Toth et al., 2018; Winker et al., 2013; Winker et al., 2009).

(4) The Raikoke eruption event is added to the comparison of instantaneously retrieved undetected aerosol extinction and Level 3 product in Section 3.3.

(5) The results, discussion and conclusions were largely rewritten in the revised manuscript to highlight the significance of this study.

**Major issues**

1. The paper requires major revisions in writing. Many sentences do not make sense, some references do not make sense within the context of the sentence, and some references do not support what the authors claim.

   *Response*: Thank you for your comment. The results, discussion and conclusions were largely rewritten in the revision. Also, the English has been carefully polished and the references have been corrected.

2. The authors do not make it clear how their method is different from previous work, which leaves me questioning the scientific merit of this work.

   *Response*: The method has been updated and is described in Section 2.4, as follows.

[revised manuscript text omitted]

3. This work is ultimately a statistical analysis and the authors are looking for faint

signal that, per their paper, is below the previously published limit of detection (LOD) for CALIPSO. Their retrieval requires an iterative process but do not attempt to propagate uncertainties through this process. Therefore, we have no was of determining whether their end product is significantly different from zero.

***Response***: Good suggestion! We have added the calculation of the extinction uncertainty for the retrieval as follows:

For the retrieved extinction of undetected aerosol, we calculated the uncertainty to assess the reliability of the results according to the algorithm of CALIPSO Level 2 aerosol product (Young et al., 2013), where the main equations are as follows:

$$\frac{\Delta\beta_N'(r)}{\beta_N'(r)} = \left\{\left[\frac{\Delta\beta'(0,r)}{\beta'(0,r)}\right]^2 + \left[\frac{\Delta C_N(r_N)}{C_N(r_N)}\right]^2\right\}^{1/2}, \tag{5}$$

$$\left(\Delta\beta_p(r)\right)^2 = \beta_T^2(r)\left[\left(\frac{\Delta\beta_N'(r)}{\beta_N'(r)}\right)^2 + \left(\frac{\Delta T_M^2(r_N,r)}{T_M^2(r_N,r)}\right)^2 + \left(\frac{\Delta T_P^2(r_N,r)}{T_P^2(r_N,r)}\right)^2\right] + \left(\Delta\beta_M(r)\right)^2, \tag{6}$$

$$\Delta\sigma_P(r) = \left[\left(\frac{\Delta S_p}{S_p}\right)^2 + \left(\frac{\Delta\beta_p(r)}{\beta_p(r)}\right)^2\right]^{1/2}\sigma_P(r), \tag{7}$$

where $\Delta\beta_p(r)$ and $\Delta\sigma_P(r)$ in Eqs (6) and (7) are the particle backscatter uncertainty and particle extinction uncertainty, respectively; they are the target parameters for the calculation. Eq. (5) is the formula for one of the terms of Eq. (6), where $\Delta\beta_N'(r)$ is the uncertainty of the renormalized TAB, $\Delta\beta'(0,r)$ is the uncertainty of the TAB, and $\Delta C_N(r_N)$ is the uncertainty of renormalization. The error due to renormalization is negligible (Kim et al., 2017) because the starting altitude of retrieval ($r_N$ =36 km) is consistent with the calibration region (36–39 km) for the CALIPSO Level 1B Version 4 product (Kar et al., 2018); therefore, $\Delta C_N(r_N)$ is set to 0. The standard deviation of the TAB is used to approximate $\Delta\beta'(0,r)$ because the TAB in this study was pre-processed.

Uncertainty is found in the calibration factor in $\Delta\beta'(0,r)$, which contains systematic and random components (Young et al., 2013), and this approximation neglects the systematic error in the calibration factor, producing a low bias in the uncertainty calculation. Fortunately, the calibration factor bias of the nighttime CALIPSO V4 product has been reduced to $1.6\% \pm 2.4\%$ (Kar et al., 2018). Additionally, Kim et al. (2017) pointed out that the bias caused by the lidar ratio is dominated in the retrieval. Thus, we consider ignoring the calibration factor in the systematic error. The other terms in Eq. (6), total backscatter coefficient ($\beta_T(r)$), molecular and particle two-way transmittance uncertainty ($\Delta T_M^2(r_N,r)$ and $\Delta T_P^2(r_N,r)$), and molecular backscatter uncertainty ($\Delta\beta_M(r)$) are calculated in the same way as in Young et al. (2013) and are not repeated here. $S_p$ and $\Delta S_p$ in Eq.

(7) are selected from the median and median absolute deviation, respectively, in the retrieved 20°×20° grid lidar ratio based on CALIPSO profile locations.

Also, we discuss the uncertainties corresponding to the different magnitudes of extinction in the retrieval, as follows:

…The averaged black line in Figure 5b show the mean relative uncertainties of CALIPSO, specifically ~35% and ~125% for the retrieved extinction of $10^{-3}$ and $10^{-4}$ km$^{-1}$, respectively.

[Figure]

Figure 5. (a) Correlation plots of the retrieval within the matching grid of CALIPSO nighttime and SAGE III/ISS product from 5 km to 30 km for 12 months of validation. The color bar represents the sample size. The black bins represent the mean values of each 10% quantile (0-10%, 10-20%...and 90-100%) of SAGE III/ISS 521 nm aerosol product and corresponding CALIPSO retrieval. The I-type bars indicate the standard error of each 10% quantile CALIPSO retrieval. (b) The relative uncertainty of one-degree CALIPSO extinction.

4. One has to wonder how valid a comparison between an instantaneous product and a monthly average is (i.e., the authors L1 product and the standard CALIPSO L3 extinction product) per their Figure 7. Further, the faint aerosol layer they identify is at ≈17.5 km and they showed the L3 product at 15.2 km. The validity of this comparison is tenuous even when the altitude match, and much less so when the altitudes do not match.

***Response***: We have displayed the height of 17 km of the CALIPSO Level 3 product background aerosol, as in Figure 8. We do the comparison between our instantaneous retrieval and the monthly CALIPSO L3 product to qualitatively show that the aerosol enhancement at tropopause and stratosphere caused by the nature (i.e. volcanic eruption) (Kloss et al., 2021; Knepp et al., 2021). A background aerosol enhancement can generally persist for a long time, so that the

instantaneous retrieved aerosol by CALIPSO can be shown at the L3 monthly-average scale. The relative revised content is as follows:

[Figure]

Figure 8. (a) and (b) are the stratospheric extinction distributions of CALIPSO Level 3 Stratospheric Aerosol Profile products at 17 km in June and August, respectively. (c) and (d) are the retrieved aerosol extinction scenes based on CALIPSO instantaneous data on June 10 and August 10, respectively, consistent with Figure 4a. The corresponding trajectories for the two scenes are shown as red lines in (a) and (b), and the corresponding aerosol subtypes are shown in (e) and (f), the same as in Figure 7c.

Figures 8a and 8b show the spatial distribution of aerosol extinction averaged in June and August 2019 at 17 km altitude from CALIPSO Level 3 monthly-averaged Stratospheric Aerosol Profile product with the resolution of 5°×20° in latitude and longitude (Kar et al., 2019). A significant amount of aerosol enhancement was observed in the stratosphere in August in the northern hemisphere (Figure 8b), possibly due to the eruption of the Raikoke Volcano in June 2019 (Knepp et al., 2021; Kloss et al., 2021). We selected two CALIPSO tracks across aerosol enhancement areas in June and August (Figures 8c and 8d), respectively. The stratosphere at the northern hemisphere latitudes is clean, whereas natural dust aerosol prevails in the lower troposphere on June 10 when Raikko has not yet erupted (Figures 8c and 8e). The clean condition shown by our retrieval is

consistent with the CALIPSO Level 3 products that indicate the clean stratosphere at a monthly temporal scale.

Following the onset of volcanic eruptions, strong stratospheric aerosol layers are found in the stratosphere between 50°N and 60°N that are classified as sulfate by the VFM (Figure 8f). As shown in the red dash box of Figure 8d, aerosol extinction enhancement (~0.005 km$^{-1}$) occurs around 17 km near 40°N to 5°N, which corresponds to the monthly average scale aerosol contamination in the stratosphere throughout the Northern Hemisphere in Figure 8b, but is not captured by CALIPSO Level 2 products (Figure 8f). Therefore, the retrieved undetected aerosol extinction can well capture the aerosol enhancement from special events at a horizontal resolution of 20 km (Figure 8d). The color ratios, particle depolarization ratios, and integrated attenuated backscatter are extracted manually for the red dashed region (16 km to 20 km, 40°N to 5°N) with an average of 0.17, 0.02 and 0.00033 sr$^{-1}$, respectively. Using these optical and non-optical properties (i.e., center height, temperature and latitude), aerosol subtypes can be determined by the CALIPSO Scene Classification Algorithms (Kim et al., 2018). The results show that the aerosol subtype in this region is sulfate, which supports that the aerosol enhancement is more likely to be from the eruption of the Raikoke Volcano.

5. It seems the authors are unaware of the 2019 Raikoke eruption, which is likely the source of aerosol in their Figs. 3, 4, 7. Instead, they attribute these layers to a single wildfire event in Siberia without providing any supportive evidence. The papers they cite in support of this claim do not, in fact, support their assertion. Ohneiser et al. 2021 claimed to see smoke up to 13 km from 50°N to 80°N. Kostrykin et al. 2021 did a modeling study wherein they claimed to trace smoke up to ≈ 6 km. The authors claim to see smoke from 10°N to 60°N at 15–20 km. This is in stark contrast to the finding of the 2 referenced works. The authors either need to support their assertion that smoke was present, and responsible for the faint aerosol layer they observed, or reconsider the source.

*Response*: You are right. We check the aerosol distribution at tropopause and stratosphere during the summer of 2019 based on CALIPSO L3 monthly-average data. The aerosol enhancements in August of 2019 are consistent with Raikoke volcanic eruption, as described in Major Comment 4. Further, we compare aerosol at 17 km in August of 2018 and 2019 (Figure R1) to confirm these aerosol enhancements are unique in 2019. The Raikoke volcanic eruption has been added to the interpretation of stratospheric aerosol enhancement in the Northern Hemisphere in August 2019.

[Figure]

Figure R1 (a) and (b) are the stratospheric extinction distributions of CALIPSO Level 3 Stratospheric Aerosol Profile products at 17 km in August 2018 and August 2019, respectively.

6. Section 3.1: The authors claim these faint aerosol layers are the product of wildfire events (i.e., smoke). Smoke has a very different lidar ratio from 50 sr$^{-1}$. Did the authors use a lidar ratio of 50 sr$^{-1}$ here? If so, how does this impact the interpretation of the results (e.g., do the faint aerosol layers disappear if the correct lidar ratio is used)?

*Response*: Yes, the lidar ratio is a crucial factor in extinction retrieval. In this study, a new method of lidar ratio in global is retrieval by using SAGE III/ISS AOD as a constraint, as described in Major Comment 2. Also, the significance of the lidar ratio obtained by this new method and the related bias are also discussed, as follows.

As mentioned in Section 2.4, the initial stratosphere and troposphere lidar ratios were derived from the empirical value (50 sr) of CALIPSO Level 3 stratospheric aerosol product (Kar et al., 2019) and the lidar ratio (28.75 sr) obtained by Kim et al. (2017), respectively. The latter is estimated from the retrieved CALIPSO column-integrated extinction with MODIS AOD constraints. As shown in Figure 9, the retrieved extinction using the fixed lidar ratio is higher than that using the SAGE-constrained lidar ratio because the median lidar ratio of the former (50 and 28.75 sr) is larger than the latter (42.2 and 24.5 sr). However, the NRMSE of retrieved extinction decreased by about 15% (from 120.2% to 105.6%) when changed the fixed lidar ratio to the SAGE-constrained lidar ratio in global. Particularly, when using the fixed lidar ratio of 50 sr in the high latitude stratosphere, it could result in a larger bias because the fixed lidar ratio is more different from the SAGE-constrained lidar ratio (~35 sr) (Figure 3a). Therefore, these indicate a better accuracy of retrieved undetected aerosol extinction using the SAGE-constrained lidar ratio in global.

[Figure]

Figure 9. The colored scatter plot is the same as that in Figure 5a, but the CALIPSO extinction are retrieved using fixed lidar ratios of 50 and 28.75 sr in stratosphere and troposphere from June 2017 to May 2020, respectively. The gray and black lines are the mean value of of each 10% quantile (as in Figure 5a) of the CALIPSO retrieved extinction using the fixed and our retrieved lidar ratios, respectively.

Figure 7b illustrates a possibly missed smoke from a wildfire. Based on the SAGE-constrained lidar ratio (median 42.2 and 24.5 sr), we retrieved and see the undetected aerosol by CALIPSO Level 2 products, which connects with two strong aerosol layers. The lidar ratio for the smoke reported in the CALIPSO Level 2 Version 4 product is $70\pm16$ sr (Young et al., 2018), which is very different from the SAGE-constrained lidar ratio for the troposphere at this location. Theoretically, a larger lidar ratio will derive a larger extinction in the retrieval. This indicates that the undetected aerosol extinction should be larger if using the smoke lidar ratio of $70\pm16$ sr. However, so for, this bias cannot be avoided here because an automatic classification is impossible when we do not know the boundaries of those aerosols. Therefore, we have to treat the stratospheric (or tropospheric) undetected aerosols as a whole and assign the same lidar ratio regardless of the aerosol type in this study. Although the retrieved extinction in Figure 7 is biased, it demonstrated the importance of retrieving high spatial-temporal resolution undetected aerosol extinction. A solution to reduce this bias is to develop a more effective layer detection and classification algorithm, and our team is already working on it (Mao et al., 2021).

**Minor issues**

1.  The authors perform their coefficient correlation and RMSE calculations in logarithmic space. Why use logarithmic space when this effectively reduces the comparison to essentially how well the orders of magnitude agree.

***Response***: In the revised manuscript, we calculated the R and NRMSE (normalized root mean square error) under linear conditions as follows:

The correlation coefficients (R) and normalized root mean square error (NRMSE) are 0.66 and 100.6%...

2. How is "faint aerosol" defined? What qualifies as a "faint aerosol"?.

   ***Response***: Sorry for the confusion. In this study, faint aerosol refers to the aerosol undetected by CALIPSO Level 2 Layer Detection Algorithm. They are too weak to be detected by CALIPSO Level 2 products. We now clarify the definition of faint aerosols and give it directly in the revised title.

   *Retrieving Instantaneous Extinction of Aerosol Undetected by CALIPSO Layer Detection Algorithm*

3. On lines 81–83 the sentence containing "...observed by the rays passing through the atmo-sphere..." does not make sense and this is not how SAGE quantifies extinction coefficient. I think, as written, this sentence is incorrect. Light passes through the atmosphere and is attenuated by some combination of scattering and absorption of molecules, particles, and clouds. The molecular number densities and extinction coefficients are then derived based on the recorded spectra.

   ***Response***: Modified according to your suggestions, as follows:

   SAGE III conducts solar and lunar occultation measurements globally while orbiting the Earth on the International Space Station (ISS). Light passes through the atmosphere and is attenuated by some combination of scattering and absorption of molecules, particles, and clouds. The extinction coefficients are then derived based on the recorded spectra (Cisewski et al., 2014; Thomason et al., 2010).

4. Throughout the paper the authors refer to SAGE extinction, but do not specify wavelength. They stated within the text that they used the 521 nm channel, but it would help the reader if the wavelength is included in all subsequent references to SAGE extinction.

   ***Response***: Thank you for your suggestion, we have highlighted 521 nm in the manuscript where SAGE aerosol products are mentioned.

5. The authors used the SAGE III/ISS v5.1 product while the v5.2 product is available. Vertical smoothing for the extinction products was turned off for the v5.2 product, which effectively provides data at a higher vertical resolution. The authors are encouraged to use this data set since the higher resolution will make their comparisons more meaningful.

   ***Response***: Thank you for your suggestion. Here, we smoothed the CALIPSO data

in the vertical direction, and the SAGE III/ISS V5.1 data with the smoothing operation is appropriate for comparison and validation. Previous studies have shown the extinction coefficients of SAGE III had a low bias with an uncertainty of around 10% (Thomason et al., 2010). Also, we have added a correction to the bias within extinction coefficients of SAGE III/ISS aerosol product due to the ozone interference, as follows:

A low bias in the extinction coefficients of the SAGE III/ISS aerosol product is observed at 521 nm due to the ozone interference in the retrieval algorithm. This finding is more pronounced at mid-latitudes and altitudes between 20 and 25 km (Knepp et al., 2021). The following equation is therefore used to correct the extinction at 521 nm (Knepp et al., 2021):

$$log\ \sigma_{521} = \frac{log\ (\frac{\sigma_{450}}{\sigma_{755}}) \times log\ (\frac{521}{755})}{log\ (\frac{450}{755})} + log\ (\sigma_{755}), \tag{1}$$

where $\sigma$ is the extinction coefficient, and the numbers represent the wavelength. We removed the bins in the SAGE III/ISS aerosol extinction profile with color ratio (the ratio of the aerosol extinction at 521 and 1022 nm) in the range of 0.8 to 1.2 to avoid cloud contamination (Schoeberl et al., 2021).

Additionally, Due to the generally large coverage of undetected faint aerosols, the vertical resolution is enough to capture undetected faint aerosols as validation. Thus, considering the limited time to revise the paper, the SAGE III/ISS V5.1 data are still used in the current study, but we will test the V5.2 data in the future.

6. Line 95 What is the "vertical moving filtering" that was done? What kind of filter is this and what is it removing?

*Response*: The "vertical moving filtering" is moving average filtering on a TAB profile using a 5-point vertical window to reduce the random noise contained in the observation. In the revised manuscript, the description of the process was rephrased in Section 2.1, as follows:

(3) The vertical resolution of the CALIPSO Level 1B TAB profiles varies with the height of 30, 60, 120, and 300 m for −0.5–8, 8–20.2, 20.2–30.1, and 30.1–40 km, respectively. Referring to Kim et al. (2017), the TAB profiles are reduced to a vertical resolution of 300 m by linear interpolation to improve the SNR, followed by a vertical moving mean filtering (with a 5-point window) and horizontal averaging to 20 km to retrieve the extinction of undetected aerosol.

7. Not all of the variables in equations 1-3 are defined within the text (some are defined within a figure, some within a figure caption, some within the text). Please

define all variables within the text.

***Response***: In the revised manuscript, we removed the flowchart (Figure 1 of the original manuscript) of the extinction retrieval algorithm, and chose to illustrate it in the text. All variables are now defined in the text, as follows:

In this study, the undetected aerosol extinction coefficient is retrieved by the Fernald method, similar to CALIPSO Level 2 and Level 3 aerosol products (Young and Vaughan, 2009; Kar et al., 2019; Young et al., 2018). Based on the pre-processed TAB (i.e., $\beta'(r)$), the particulate backscatter coefficient (i.e., $\beta_p(r)$) is solved by iterating Eqs. (2) and (3c) in the following equations:

$$\beta_p(r) = \frac{\beta'(r)}{T_m^2(r)T_{o_3}^2(r)T_p^2(r)} - \beta_m(r), \tag{2}$$

$$T_m^2(r) = exp\left(-2\int_0^r \alpha_m(r')\,dr'\right), \tag{3a}$$

$$T_{o_3}^2(r) = exp\left(-2\int_0^r \alpha_{o_3}(r')\,dr'\right), \tag{3b}$$

$$T_p^2(r) = exp\left(-2\eta_p S_p \int_0^r \beta_p(r')\,dr'\right), \tag{3c}$$

$$\alpha_p(r) = S_p(r)\beta_p(r), \tag{4}$$

where $T_m^2(r)$, $T_{o_3}^2(r)$, and $T_p^2(r)$ represent the molecular, ozone, and particulate two-way transmittances, respectively. The molecular backscatter coefficients ($\beta_m(r)$) and molecular and ozone two-way transmittances ($T_m^2(r)$ and $T_{o_3}^2(r)$) can be calculated from the molecular number density and ozone number density provided by CALIPSO Level 1B product, respectively. The $\alpha_m(r)$, $\alpha_{o_3}(r)$ and $\alpha_p(r)$ represent the extinction coefficient of molecular, ozone, and particulate, respectively. The retrieval algorithm has several basic settings. The multiple scattering coefficient ($\eta_p$) for undetected aerosol particles is set to 1 as considered in the retrieval of the CALIPSO Level 2 product (Young et al., 2018). Meanwhile, the bin at 36 km is considered aerosol-free (i.e., $\beta_p(0) = 0$, $T_p^2(0) = 1$) (Kar et al., 2019).

8. Lines 120–121: Please provide reference for β(0) = 0, Tp(0) = 1

***Response***: Provided, as follows:

Meanwhile, the bin at 36 km is considered aerosol-free (i.e., $\beta_p(0) = 0$, $T_p^2(0) = 1$) (Kar et al., 2019).

9.  Line 124: What is meant by "same day"? Is this the same calendar date, or within 24 hours?

    ***Response***: Sorry for the confusion. The "same day" refers to the same calendar date, which has been noted in the manuscript now.

    Since only daytime data from SAGE III/ISS are available, the CALIPSO orbits are spatially and temporally matched to the nearest SAGE III/ISS observations on the same calendar date with the consideration for a smaller temporal-spatial variation of faint aerosol comparing strong aerosol at the near-surface.

10. Line 125: You state the SAGE horizontal resolution is ≈300 km; please provide a reference for this value. This raises a greater point: the viewing geometries and sampling volumes of SAGE and CALIPSO are vastly different and may provide another source of uncertainty.

    ***Response***: This value (300 km) comes from the SAGE official website of https://space.oscar.wmo.int/instruments/view/sage_iii. The 2°×1° grid is used to match CALIPSO and SAGE III/ISS. All profiles of CALIPSO within a 2°×1° grid are used for extinction retrieval averagely to minimize the uncertainty. The detail about how to match CALIPSO and SAGE III/ISS have been revised in Section 2.3, as follows:

    Since only daytime data from SAGE III/ISS are available, the CALIPSO orbits are spatially and temporally matched to the nearest SAGE III/ISS observations on the same calendar date with the consideration for a smaller temporal-spatial variation of faint aerosol comparing strong aerosol at the near-surface. The horizontal resolution of SAGE III/ISS occultation observations is low with ~300 km (https://space.oscar.wmo.int/instruments/view/sage_iii). Thus, we selected a 2° × 1° (longitude × latitude) grid centered on the SAGE III/ISS observations to match CALIPSO instantaneous observation. To ensure enough CALIPSO profiles are included for each successfully matched sample, the CALIPSO track crossed the grid and have to exceed 0.75° latitude (Figure 1a). The 2° longitude is to obtain the successfully matched samples as soon as possible. Figure 1b shows the global distribution of nighttime CALIPSO and SAGE III/ISS match numbers in 20°×20° grids for three years from June 2017 to May 2020. No successful match in the grids is found in the black boundary due to the removal of low-energy laser shots of CALIPSO in the SAA region. Finally, 1349 and 1325 profiles are successfully matched for CALIPSO nighttime and daytime data with SAGE, respectively.

[Figure]

Figure 1. (a) Schematic of the CALIPSO match to the SAGE III/ISS. The red circle represents the center point of SAGE III/ISS observations. The red and blue lines represent cases of successful and failed (CALIPSO track is less than 0.75° in the grid) matches, respectively. (b) Global matches of nighttime CALIPSO and SAGE III/ISS from June 2017 to May 2020. The black boundary represents the South Atlantic Anomaly (SAA), where CALIPSO is experiencing an elevated frequency of low-energy laser shots. The color bar represents the number of matches in each 20°×20° grid.

Also, the results in Section 3.2 show a case of the comparison after matching and averaging, as follows. It shows good agreement between CALIPSO and SAGE.

Figure 4a shows a case of the retrieved CALIPSO extinction at latitude 33° on August 26, 2019. An undetected faint aerosol layer (extinction coefficients around 0.005 km$^{-1}$) is connected to the detected stratospheric aerosol layer provided by the CALIPSO Level 2 aerosol product at altitudes of 15 km to 20 km around 10°N to 40°N latitude. Figure 4b shows high-consistent extinctions of CALIPSO undetected aerosol and matched SAGE III/ISS 521 nm aerosol product (red dash line in Figure 4a) above 15 km. Additionally, this profile comparison demonstrates the feasibility of ignoring the diurnal variation of undetected aerosols.

[Figure]

Figure 4. (a) Latitude-altitude undetected aerosol extinction based on CALIPSO nighttime data on August 26, 2019. The color represents the extinction coefficient (km$^{-1}$). The purple and black boundary lines represent the detected aerosol and

cloud layers provided by CALIPSO Level 2 products, respectively. The gray line represents tropospheric height. The red dash line is the observation position of SAGE III/ISS. The white areas represent the removed data inside and below the detected layers. The retrieved faint aerosol at 20 km is shown after additional mean filtering (3×3 window) to highlight the faint aerosol area. (b) Comparison of faint aerosol extinction (km$^{-1}$) profile for matched CALIPSO and SAGE III/ISS 521 nm aerosol product. The gray lines represent the undetected aerosol extinction of CALIPSO retrieval at a resolution of 20 km horizontally and 0.3 km vertically, and the blue line represents averaged gray lines. The red line represents the aerosol extinction from SAGE III/ISS.

11. Line 126: The reference to Adams et al. 2013 does not make sense here. Please include why this is relevant or remove the reference.

    ***Response***: This reference has been removed.

12. Line 127: What is meant by "...where the CALIPSO orbit exceeds 0.75° latitude..."? What is meant by 0.75° ? This is confusing, please clarify.

    ***Response***: Sorry for the confusion. The 0.75° represents the minimum length threshold of CALIPSO at a successfully matched sample. The sentence has been rephrased as described in in the response of Minor Comment 10.

13. Line 144: The authors refer to a "red dashed box" in Fig. 3 (a), but there is no such box in that figure. Please add the box.

    ***Response***: Modified, as follows.

[Figure]

Figure 7. (a) MODIS Terra true-color image in the daytime and the passing CALIPSO track (yellow line) at night. (b) Latitude–altitude aerosol extinction of the corresponding nighttime CALIPSO track, same as in Figure 4a. The purple and

black boundary lines and extinction inside represent the detected aerosol and cloud layers provided by CALIPSO Level 2 products, respectively. (c) Aerosol subtypes in CALIPSO VFM product (N/A=not applicable, 1=marine, 2=dust, 3=polluted continental/smoke, 4=clean continental, 5=polluted dust, 6=elevated smoke, 7=dusty marine, 8=PSC aerosol, 9=volcanic ash, 10=sulfate/other). (d) Attenuated scattering ratio.

14. Lines 148–150: One has to ask, if these faint aerosol layers are visible within the ASR product, then why go to the trouble of calculating extinction coefficient? It seems that computing extinction coefficient introduces unnecessary uncertainties, so my 2 questions are: 1. why use the extinction coefficient?, 2. are the uncertainties that the extinction calculation introduced acceptable?

*Response*: Thank you for your questions.

(1) The ASR is a kind of lidar signal which is not a physical parameter that can be used directly in scientific research. For instance, the calculation of radiative effects of aerosols requires extinction as an input, not ASR.

(2) We have added the calculation of the extinction uncertainty for the retrieval as described in Major Comment 3. The comparison between CALIPSO and SAGE shows good agreement with the independent SAGE aerosol extinction (R=0.66). Especially, the relative uncertainties of the retrieved extinction coefficients at $10^{-3}$ km$^{-1}$ and $10^{-4}$ km$^{-1}$ are 35% and 125%, respectively, while the minimum extinction of CALIPSO L2 is 0.01 km$^{-1}$ with 40% uncertainty (Kacenelenbogen et al., 2011; Toth et al., 2018; Winker et al., 2013; Winker et al., 2009).

15. Line 156: It seems the authors are introducing an additional step in their processing algorithm (i.e., the 3x3 mean filtering window). This should be included in the main body of the text and explain what this is.

*Response*: The 3×3 window filter is not used in the retrieval, which is only used to highlight the areas of aerosol enhancement in the mapping of the figure. We have rephrased the figure title.

16. Line 163: Perhaps the pdf did not build properly, but there is no red dash line in Fig. 4 (a) though there is a pink line.

*Response*: Modified, as described in the response of Minor Comment 15.

17. Figure 4: What is the vertical pink dashed line?

*Response*: The pink dashed line (red dashed line in the revised manuscript) indicates the location of the SAGE III/ISS observation, as described in the response

of Minor Comment 15.

18. Line 172 and Figure 5 (a): How were these averages calculated? There are 11 points on this line, but the SAGE data exists over a continuum, so there must have been some binning along the x-axis.

*Response*: You are right, there are some binning along the X-axis (SAGE III/ISS data). In the revised manuscript, the binning ranges are 0-10%, 10-20% … 90-100%, for a total of 10 points, as follows.

[Figure]

Figure 5. (a) Correlation plots of the retrieval within the matching grid of CALIPSO nighttime and SAGE III/ISS product from 5 km to 30 km for 12 months of validation. The color bar represents the sample size. The black bins represent the mean values of each 10% quantile (0-10%, 10-20%...and 90-100%) of SAGE III/ISS 521 nm aerosol product and corresponding CALIPSO retrieval. The I-type bars indicate the standard error of each 10% quantile CALIPSO retrieval. (b) The relative uncertainty of one-degree CALIPSO extinction.

19. Line 172 and Figure 5 (a): The authors refer to this as "faint aerosol extinction", but does this really qualify as "faint" when the SAGE extinction coefficients exceed 1E-3 (even 1E-2!)?

*Response*: Thank you for the question. In this study, faint aerosol refers to the aerosol undetected by CALIPSO Layer Detection Algorithm. They are too weak to be detected by CALIPSO Level 2 products. We now clarify the definition of faint aerosols and give it directly in the revised title.

*Retrieving Instantaneous Extinction of Aerosol Undetected by CALIPSO Layer Detection Algorithm*

20. Line 173: The authors claim the CALIPSO retrieval has a low bias while their line of best fit falls just above the 1:1 line, indicating the CALIPSO retrieval has a high

bias. Can the authors explain what they mean by low bias and justify this designation?

***Response***: Sorry for the confusion. It has now been rephrased for Figure 5 as follows.

The nighttime CALIPSO undetected aerosol extinction and SAGE III/ISS 521 nm aerosol extinction show good agreement for the 12-month validation dataset (Figure 5a), with the average retrieved aerosol extinction (black line) closing to the 1:1 line. The correlation coefficients (R) and normalized root mean square error (NRMSE) are 0.66 and 100.6% based on the independent 12-month SAGE validation dataset, respectively. ...

…The averaged black line in Figure 5b show the mean relative uncertainties of CALIPSO, specifically ~35% and ~125% for the retrieved extinction of $10^{-3}$ and $10^{-4}$ km$^{-1}$, respectively.

[Figure]

Figure 5. (a) Correlation plots of the retrieval within the matching grid of CALIPSO nighttime and SAGE III/ISS product from 5 km to 30 km for 12 months of validation. The color bar represents the sample size. The black bins represent the mean values of each 10% quantile (0-10%, 10-20%...and 90-100%) of SAGE III/ISS 521 nm aerosol product and corresponding CALIPSO retrieval. The I-type bars indicate the standard error of each 10% quantile CALIPSO retrieval. (b) The relative uncertainty of one-degree CALIPSO extinction.

21. Line 184: What is meant by "mean values of 5% quantiles"? It seems the authors have introduced another calculation that is not defined/explained within the text and is not readily understood by the reader. This should be explained within the text.

***Response***: Sorry for the confusion. The confusion has been removed as described in the response of Minor Comment 18.

22. Line 193: The authors state "...the CALIPSO retrievals in the troposphere should be more reliable [than SAGE]" While I agree about this statement to some degree, this is a blanket statement that is made without qualifier and is misleading. The authors need to explain what they mean here. Further, there are nuances that are important. In the troposphere you can have highly variable lidar ratios; how does this influence the "reliability"? You have to propagate your errors all the way through the stratosphere before hitting the troposphere; how does this influence reliability? Finally, the troposphere is often loaded with aerosol (as compared to the stratosphere), so how often do you expect to be below the $0.01\ \mathrm{km}^{-1}$ LOD (i.e., is you method still necessary at this point)? It seems like the discussion regarding the troposphere is misplaced within this paper.

***Response***: We are very grateful for your careful review. We have made the following revisions in response to this comment.

(1) Removed the section of comparing the vertical distribution of CALIPSO and SAGE signal quality, because this is not the focus of our study.

(2) Added the calculation and discussion about the uncertainty of CALIPSO extinction retrieval, as the response to Major Comment 3.

(3) We redefine the faint aerosol as the undetected aerosol by CALIPSO Layer Detection Algorithm, as we answered in Minor Comment 19. And undetected aerosol also often exists in the troposphere, so our retrieval is still necessary for the troposphere.

23. Line 201: What is the purpose of the Kim et al 2017 reference here?

***Response***: This citation is intended to illustrate that the typical SNR ratio at 35 km of 20-km daytime CALIPSO data is less than 1 calculated by Kim et al. (2017), which is similar to our results. It has been removed from the revised manuscript due to the large revisions of this paper.

24. Section 3.3: The authors attribute the faint aerosol during this time period to the Siberian wildfires. As discussed above this is more likely the impact of the Raikoke eruption. It is strange that this was not mentioned here. Please see discussion above regarding this topic and address the issues stated therein.

***Response***: Sorry for the confusion. We check the aerosol distribution at tropopause and stratosphere during the summer of 2019 based on CALIPSO L3 monthly-average data. The aerosol enhancements in August of 2019 are consistent with Raikoke volcanic eruption, as described in the response of Major Comment 4. Further, we compare aerosol at 17 km in August of 2018 and 2019 (Figure R1) to confirm these aerosol enhancements are unique in 2019. The Raikoke volcanic

eruption has been added to the interpretation of stratospheric aerosol enhancement in the Northern Hemisphere in August 2019 (Kloss et al., 2021; Knepp et al., 2021), as described in the response of Major Comment 4.

[Figure]

Figure R1 (a) and (b) are the stratospheric extinction distributions of CALIPSO Level 3 Stratospheric Aerosol Profile products at 17 km in August 2018 and August 2019, respectively.

25. Figure 7: It will help the reader if panels (a), (b), (c), and (d) all have the same colorbar scale and limits.

    ***Response***: Thank you for your suggestion. Figures 7c-7d (Figure 8c-8d in the revised manuscript) represent the undetected aerosol extinction of a CALIPSO track from surface to 30 km, with values varying over a wide range. If we use the same scale and limits of the colorbar in Figures 7a-b as in Figures 7c-d, we will lose lots of details. Particularly, Figures 7a will be all in blue.

26. Line 214: The Kostrykin reference in not germane to this paper as they showed no indication of the Siberian wildfire smoke leaving the troposphere. They estimated a 5% chance of smoke making it to 6 km.

    ***Response***: Thank you for pointing out this problem, the reference has been removed.

27. Line 224: "...classified as elevated smoke by VFM). This is categorically wrong. The CALIPSO vertical feature mask identifies this plume as "sulfate/other" not smoke.

    ***Response***: You are right, it has been modified.

28. Lines 227–228 and Figure 7: It is difficult to compare a monthly average and an "instantaneous" measurement. However, the layer referenced in Fig. 7 (d) is not at the same altitude as what is shown in Fig. 7 (b). Going off Fig. 7 (b) alone, I would expect little (no?) enhancement at 15.2 km for most of the the NH. If we accept Fig. 7 in its current state we cannot draw a decisive conclusion (panels (b) and (d) contradict each other). However, if panels (a, b) were updated to mate the altitude of the layer in (d) this would significantly bolster your case.

***Response***: We have displayed the 17 km background aerosol of the CALIPSO Level 3 product, as shown in Figure 8. We do the comparison between our instantaneous retrieval and the monthly CALIPSO L3 product to qualitatively show that the aerosol enhancement at tropopause and stratosphere is caused by the nature (i.e. volcanic eruption) (Kloss et al., 2021; Knepp et al., 2021). A background aerosol enhancement can generally persist for a long time, so that the instantaneous retrieved aerosol by CALIPSO can be shown at the L3 monthly-average scale, as described in the response of Major Comment 4.

[Figure]

Figure 8. (a) and (b) are the stratospheric extinction distributions of CALIPSO Level 3 Stratospheric Aerosol Profile products at 17 km in June and August, respectively. (c) and (d) are the retrieved aerosol extinction scenes based on CALIPSO instantaneous data on June 10 and August 10, respectively, consistent with Figure 4a. The corresponding trajectories for the two scenes are shown as red lines in (a) and (b), and the corresponding aerosol subtypes are shown in (e) and (f), the same as in Figure 7c.

29. Line2 242–246, conclusion #1: As stated above, some of your fundamental assumptions will be different within the troposphere. Further, the discussion of tropospheric aerosols does not seem to fit within the context of this paper. Further, the authors failed to demonstrate that this method is novel from previous work.

Therefore I find this conclusion to be tenuous.

*Response*: Thank you for your comment. Based on the new methods, results and discussions in the revision, we rewrite the conclusions as follows:

(1) The lidar ratio for the stratosphere and troposphere in global is derived based on CALIPSO instantaneous observations using SAGE III/ISS AOD as a constraint. The derived lidar ratio is significantly higher in the stratosphere (median 42.2 sr) than that in the troposphere (median 24.5 sr). The derived lidar ratio peak at the equator and decrease with latitude at the stratosphere, while the lidar ratio variations are small at the troposphere in global.

(2) The retrieved undetected aerosol extinction based on CALIPSO nighttime instantaneous observations shows good agreement with the SAGE III/ISS product on a 1° average. The correlation (R) and NRMSE are 0.66 and 100.6% based on the independent 12-months SAGE III/ISS data, respectively. The uncertainties of retrieved extinction coefficients at $10^{-3}$ km$^{-1}$ and $10^{-4}$ km$^{-1}$ are ~35% and 125% during nighttime, respectively.

(3) The comparison of retrieved undetected aerosol extinction based on globally fixed and SAGE-constrained lidar ratios indicates the NRMSE decreased by about 15% (from 120.2% to 105.6%) during nighttime. Additionally, the CALIPSO retrieval during daytime has a positive bias and relatively low agreement with SAGE III/ISS; it exhibits R and NRMSE of 0.25 and 454.5%, respectively, due to the low signal-to-noise ratio caused by sunlight.

(4) In the case of the Australian wildfire event, instantaneous retrieved extinction of missed aerosol from CALIPSO Level 2 products provides more details of aerosol distribution. In addition, compared with the CALIPSO Level 3 stratospheric aerosol product, the retrievals show consistent aerosol enhancement possibly due to the eruption of Raikoke Volcano, but at a higher spatial-temporal resolution.

30. Lines 247–248, conclusion #2: While the authors demonstrated this they were not the first to determine this so I do not view this as a significant finding/conclusion to be drawn from this paper.

*Response*: This conclusion has been removed.

31. Lines 249–252, conclusion #3: While the authors demonstrated a faint aerosol layer that extended from 10◦N to 60◦N (at 20 km) the wrongly classify this as smoke. Further, they claimed the corresponding k532 was significantly below the CALIPSO L2 LOD (LOD is 0.01 km−1, their aerosol layer was ≈0.003 km−1) and they failed to account for the propagation of error in their extinction coefficient

retrieval. Therefore, we do not know if the layer they identified is significantly different from background. While I suspect this layer is real, it is difficult to interpret the significance of this conclusion without having this thorough statistical support.

*Response*: Thank you for your comment. We have added the calculation of the extinction uncertainty for the retrieval to support our conclusion, as described in the response of Major Comment 3.

**Working issues**

1. Line 13: " the susceptibility of clouds to aerosols is more pronounced when the aerosols are faint." This sentence does not make sense. Please clarify meaning.

   *Response*: The related sentences in the Abstract have been revised to remove confusion, as follows:

   Particularly, the susceptibility of cloud and precipitation to aerosols are stronger when aerosols are faint, but tend to be saturated in polluted conditions. However, previous methodologies generally miss these faint aerosols based on instantaneous observations because they are extremely optically thin to be detected and thereby usually un-retrieved. This result in a large underestimation when quantifying aerosol climate impacts.

2. Line 39: What is the "undetected tenuous aerosol layer"? This needs defined.

   *Response*: In this study, faint aerosol refers to the aerosol undetected by CALIPSO Layer Detection Algorithm. We now clarify the definition of faint aerosols and give it directly in the revised title.

   *Retrieving Instantaneous Extinction of Aerosol Undetected by CALIPSO Layer Detection Algorithm*

3. Line 48: The word "tenuous" does not make sense here. Please rephrase.

   *Response*: The "tenuous" have been revised to "faint".

4. Lines 60–63: This sentence does not make sense. Are the "faint aerosols" composed of both background AND undetected aerosol layers? If so, what does it mean for a particle to be composed of undetected aerosol layers?

   *Response*: Sorry for the confusion. In this study, faint aerosol refers to the aerosol undetected by CALIPSO Layer Detection Algorithm. We now clarify the definition of faint aerosols and give it directly in the revised title.

   *Retrieving Instantaneous Extinction of Aerosol Undetected by CALIPSO Layer Detection Algorithm*

5. Lines 83–84: You discuss the extinction coefficient uncertainties of SAGE III/ISS and cite Thomason et al. 2010. The Thomason paper is in reference to SAGE III/METEOR (or SAGE III/M3M). Please indicate that this reference is for SAGE III/M3M and not SAGE III/ISS.

   *Response*: Thanks for your suggestion. This paragraph has been reorganized and this sentence has been deleted.

6. Please update all "SAGE III-ISS" to "SAGE III/ISS".

   *Response*: Modified.

7. I had to read the first paragraph of section 2.3 several times before I understood your methodology. It will help the reader if you change the last sentence of this paragraph to: "To improve the signal to noise ratio (SNR) of the total attenuated backscatter (TAB) data, we performed the following pre-processing steps prior to running the data through the algorithm presented in Fig. 1:"

   *Response*: Sorry for the confusion. For the methodology section we have reorganized, we put the data pre-processing steps in section 2.1 as follows.

   2.1 CALIPSO data and pre-processing

[revised manuscript text omitted]

8. Line 102: "particle particulate multiple scattering factor" is called "multiple scattering coefficient" at other places in the manuscript. Please make this consistent throughout.

    ***Response***: Modified as you suggested.

9. Line 124: "diurnal variation of faint aerosols in SAGE III-ISS data" does not make sense. Perhaps you meant "...dirunal variation of faint aerosol within the stratosphere"?

    ***Response***: Modified, as follows:

    Since only daytime data from SAGE III/ISS are available, the CALIPSO orbits are spatially and temporally matched to the nearest SAGE III/ISS observations on the same calendar date with the consideration for a  smaller temporal-spatial variation of faint aerosol comparing strong aerosol at the near-surface.

10. Figure 5 caption: Much of this caption is difficult to understand and should be rewritten for clarity. Some information should be removed and put in the text and the authors should organize the caption in such a way as to explain what each panel is in a successive manner. In its current state the caption explains panels (a), (b), (c), then returns to discussion panel (a) without telling the reader what is going on.

    ***Response***: Thanks to your suggestion, Figure 5 has been redrawn in the revised manuscript and the caption has been rewritten as follows:

[Figure]

Figure 5. (a) Correlation plots of the retrieval within the matching grid of CALIPSO nighttime and SAGE III/ISS product from 5 km to 30 km for 12 months of validation. The color bar represents the sample size. The black bins represent the mean values of each 10% quantile (0-10%, 10-20%...and 90-100%) of SAGE III/ISS 521 nm aerosol product and corresponding CALIPSO retrieval. The I-type bars indicate the standard error of each 10% quantile CALIPSO retrieval. (b) The relative uncertainty of one-degree CALIPSO extinction.

11. Line 190: This sentence ("The number of matched points...") does not make sense. Please reword this to better communicate what you did.

   *Response*: Thanks for your suggestion but this paragraph has been reorganized and this sentence has been deleted.

12. Lines 194–195: This last sentence does not make sense (what is un-performed?). Please rewrite this to better communicate what you intend.

   *Response*: Thanks for your suggestion but this paragraph has been reorganized and this sentence has been deleted.

13. Line 210: Please include the year (e.g., June and August 2019).

   *Response*: Modified as you suggested.

14. Lines 229–230: This sentence is difficult to understand. What is a "propagation trajectory" and how can you estimate that based on a single CALIPSO granule? Please rewrite this sentence to better explain what you mean.

   *Response*: Thank you for the suggestion, this sentence has been reworded, as follows:

[revised manuscript text omitted]

---

## Author Comment (AC4)

**Manuscript ID:** acp-2022-56

**Original title:** CALIPSO Retrieval of Instantaneous Faint Aerosol

**Revised title:** Retrieving Instantaneous Extinction of Aerosol Undetected by CALIPSO Layer Detection Algorithm

The analysis presented in the manuscript demonstrates the ability to detect "faint" aerosol that is unreported in CALIPSO level 2 retrievals because it lies below feature detection thresholds. The manuscript explains the importance of quantifying this under-represented aerosol based on literature. In order to "detect" the missing aerosol, the authors follow a similar procedure as is used to construct the CALIPSO level 3 stratospheric aerosol product (Kar et al., 2019): cloud and aerosol layers detected by CALIOP are removed from the level 1 attenuated backscatter and then a Fernald retrieval is performed using a fixed lidar ratio. The difference is that the CALIPSO level 3 product reports monthly averages of aerosol extinction, whereas this manuscript analyzes extinction retrieved from individual level 2 granules. This is what is meant by the "instantaneous" descriptor.

The manuscript shows three examples where the CALIOP level 2 algorithms did not detect aerosol layers, but the extinction retrieved by the authors does indicate aerosol enhancement. It goes on to compare their retrieved aerosol extinction values to co-located SAGE III measurements, finding decent correlation at night and a high-bias in their CALIPSO retrievals during the daytime. The logic here is that because their retrieved aerosol extinction at night matches well with SAGE III measurements, then the retrieved aerosol extinction is a fair representation of what was missed by CALIOP level 2 feature detection.

*Response*: Thank you for your careful review. We have made great efforts to address the issues you raised, and the manuscript has been largely improved. The main changes are as follows.

(1) The title of the paper has been revised as "*Retrieving Instantaneous Extinction of Aerosol Undetected by CALIPSO Layer Detection Algorithm*" to highlight the objectives of the study.

(2) A new method to retrieve lidar ratios by using SAGE III/ISS products as a constraint has been added to the manuscript (see more details in Section 2.4). This study no longer uses the fixed lidar ratio (i.e. 28.75 sr) as in previous study of Kim et al. (2017). We also add a comparison of retrieved undetected aerosol extinction based on globally SAGE-constrained and fixed lidar ratios in Section 3.4 to highlight the effect of the lidar ratio we updated.

(3) Uncertainties in the extinction coefficient retrieval were calculated to assess the

reliability of the extinction results of undetected aerosol. The comparison shows good agreement with the independent SAGE III/ISS aerosol extinction (R=0.66). Especially, the relative uncertainties of the retrieved extinction coefficients at $10^{-3}$ $km^{-1}$ and $10^{-4}$ $km^{-1}$ are 35% and 125%, respectively, while the minimum extinction of CALIPSO L2 product is 0.01 $km^{-1}$ with 40% uncertainty (Kacenelenbogen et al., 2011; Toth et al., 2018; Winker et al., 2013; Winker et al., 2009).

(4) The Raikoke eruption event is added to the comparison of instantaneously retrieved undetected aerosol extinction and Level 3 product in Section 3.3.

(5) The results, discussion and conclusions were largely rewritten in the revised manuscript to highlight the significance of this study.

**General comments**

1. There are some areas where greater details are needed to avoid confusing readers. For example, the manuscript discusses "detectable extinction" by CALIOP multiple times. This is inaccurate because the CALIOP level 2 algorithms do not detect extinction. They detect aerosol layers using attenuated scattering ratio and then perform an extinction retrieval. The minimum "detectable" extinction is really just the minimum extinction occurring within detected aerosol layers. This distinction is important and needs to be made clear. Based on this and my specific comment below about the lack of details regarding CALIPSO vertical resolution, I recommend that the authors provide more information about the details of the CALIPSO level 1 product, and the steps involved in how the level 2 algorithms ultimately retrieve extinction. That would provide important context for the reader.

*Response*: Thank you for your suggestion. The expression "detectable extinction" has been corrected as you suggested in the revision. Furthermore, a description of the vertical resolution of CALIPSO Level 1 products has been added in Section 2.1, as follows:

(3) The vertical resolution of the CALIPSO Level 1B TAB profiles varies with the height of 30, 60, 120, and 300 m for −0.5–8, 8–20.2, 20.2–30.1, and 30.1–40 km, respectively. Referring to Kim et al. (2017), the TAB profiles are reduced to a vertical resolution of 300 m by linear interpolation to improve the SNR, followed by a vertical moving mean filtering (with a 5-point window) and horizontal averaging to 20 km to retrieve the extinction of undetected aerosol.

The technical process for CALIPSO Level 2 products has also been described, as follows.

The CALIPSO mission introduced new technology for retrieving aerosol profiles

from space since April 2006, with a dual-wavelength backscattering lidar as the primary payload (Winker et al., 2010). The CALIPSO team has released different levels of products for different scientific objectives. Level 1 products are calibrated observations containing environmental parameters. Level 2 products are physical, chemical, and optical parameters of aerosol layers and cloud layers obtained according to a series of technical routes. The aerosol and cloud layers are firstly detected by the Selective Iterative Boundary Locator (SIBYL) (Vaughan et al., 2009), then classified by the Scene Classification Algorithms (SCA) (Kim et al., 2018), and finally the extinction coefficient is retrieved according to the Hybrid Extinction Retrieval Algorithm (Winker et al., 2010; Young et al., 2018). Level 3 products provide monthly averaged gridded global distribution data of clouds and aerosols (Kar et al., 2019).

2. The impact of lidar ratio selection is important and inadequately discussed in the manuscript. Two values are used for this analysis (50 sr stratospheric, 28.75 sr tropospheric). The manuscript justifies the two selections for generalized values. However, the aerosol type is known in at least two of the specific cases evaluated: smoke. Since smoke lidar ratios are around 70 sr, this leads to a sizable bias. The manuscript should add a discussion of the limitations of the lidar ratios used by the method.

*Response*: Yes, the lidar ratio selection is one of the keys in retrieval. We think about this question very carefully, and develop a new method to retrieve lidar ratios by using SAGE III/ISS products as a constraint, as described in Section 2.4 and follows.

[revised manuscript text omitted]

Figure 7b illustrates a possibly missed smoke from a wildfire. Based on the SAGE-constrained lidar ratio (median 42.2 and 24.5 sr), we retrieved and see the undetected aerosol by CALIPSO Level 2 products, which connects with two strong aerosol layers. The lidar ratio for the smoke reported in the CALIPSO Level 2 Version 4 product is $70\pm16$ sr (Young et al., 2018), which is very different from the SAGE-constrained lidar ratio for the troposphere at this location. Theoretically, a larger lidar ratio will derive a larger extinction in the retrieval. This indicates that the undetected aerosol extinction should be larger if using the smoke lidar ratio of 70±16 sr. However, so for, this bias cannot be avoided here because an automatic classification is impossible when we do not know the boundaries of those aerosols. Therefore, we have to treat the stratospheric (or tropospheric) undetected aerosols as a whole and assign the same lidar ratio regardless of the aerosol type in this study. Although the retrieved extinction in Figure 7 is biased, it demonstrated the importance of retrieving high spatial-temporal resolution undetected aerosol extinction. A solution to reduce this bias is to develop a more effective layer detection and classification algorithm, and our team is already working on it (Mao et al., 2021).

3.  The conclusions claim to be able to retrieve aerosol extinction down to 0.0001 /km. At that level, however, it is important to consider uncertainty and biases. It is not discussed how large the relative uncertainties are for such small extinction values for the averaging being used. A greater discussion on uncertainties should be added to specify the value the proposed method yields to capturing undetected aerosol.

*Response*: Thank you very much for your suggestion. To make our conclusion more clear and confident, we have added the calculation of the extinction uncertainty for the retrieval in Section 2.4, as follows:

For the retrieved extinction of undetected aerosol, we calculated the uncertainty to assess the reliability of the results according to the algorithm of CALIPSO Level 2 aerosol product (Young et al., 2013), where the main equations are as follows:

$$\frac{\Delta\beta'_N(r)}{\beta'_N(r)} = \left\{\left[\frac{\Delta\beta'(0,r)}{\beta'(0,r)}\right]^2 + \left[\frac{\Delta C_N(r_N)}{C_N(r_N)}\right]^2\right\}^{1/2}, \tag{5}$$

$$\left(\Delta\beta_p(r)\right)^2 = \beta_T^2(r)\left[\left(\frac{\Delta\beta'_N(r)}{\beta'_N(r)}\right)^2 + \left(\frac{\Delta T_M^2(r_N,r)}{T_M^2(r_N,r)}\right)^2 + \left(\frac{\Delta T_P^2(r_N,r)}{T_P^2(r_N,r)}\right)^2\right] + \left(\Delta\beta_M(r)\right)^2, \tag{6}$$

$$\Delta\sigma_P(r) = \left[ \left(\frac{\Delta S_p}{S_p}\right)^2 + \left(\frac{\Delta\beta_p(r)}{\beta_p(r)}\right)^2 \right]^{1/2} \sigma_P(r) , \tag{7}$$

where $\Delta\beta_p(r)$ and $\Delta\sigma_P(r)$ in Eqs (6) and (7) are the particle backscatter uncertainty and particle extinction uncertainty, respectively; they are the target parameters for the calculation. Eq. (5) is the formula for one of the terms of Eq. (6), where $\Delta\beta'_N(r)$ is the uncertainty of the renormalized TAB, $\Delta\beta'(0,r)$ is the uncertainty of the TAB, and $\Delta C_N(r_N)$ is the uncertainty of renormalization. The error due to renormalization is negligible (Kim et al., 2017) because the starting altitude of retrieval ($r_N$ =36 km) is consistent with the calibration region (36–39 km) for the CALIPSO Level 1B Version 4 product (Kar et al., 2018); therefore, $\Delta C_N(r_N)$ is set to 0. The standard deviation of the TAB is used to approximate $\Delta\beta'(0,r)$ because the TAB in this study was pre-processed.

Uncertainty is found in the calibration factor in $\Delta\beta'(0,r)$, which contains systematic and random components (Young et al., 2013), and this approximation neglects the systematic error in the calibration factor, producing a low bias in the uncertainty calculation. Fortunately, the calibration factor bias of the nighttime CALIPSO V4 product has been reduced to 1.6%$\pm$2.4% (Kar et al., 2018). Additionally, Kim et al. (2017) pointed out that the bias caused by the lidar ratio is dominated in the retrieval. Thus, we consider ignoring the calibration factor in the systematic error. The other terms in Eq. (6), total backscatter coefficient ($\beta_T(r)$), molecular and particle two-way transmittance uncertainty ($\Delta T_M^2(r_N,r)$ and $\Delta T_P^2(r_N,r)$), and molecular backscatter uncertainty ($\Delta\beta_M(r)$) are calculated in the same way as in Young et al. (2013) and are not repeated here. $S_p$ and $\Delta S_p$ in Eq. (7) are selected from the median and median absolute deviation, respectively, in the retrieved 20°×20° grid lidar ratio based on CALIPSO profile locations.

Also, we discuss the uncertainties corresponding to the different magnitudes of extinction in the retrieval in Section 3.2, as follows:

The nighttime CALIPSO undetected aerosol extinction and SAGE III/ISS 521 nm aerosol extinction show good agreement for the 12-month validation dataset (Figure 5a), with the average retrieved aerosol extinction (black line) closing to the 1:1 line. The correlation coefficients (R) and normalized root mean square error (NRMSE) are 0.66 and 100.6% based on the independent 12-month SAGE validation dataset, respectively….

…The averaged black line in Figure 5b show the mean relative uncertainties of CALIPSO, specifically ~35% and ~125% for the retrieved extinction of $10^{-3}$ and $10^{-4}$ km$^{-1}$, respectively.

[Figure]

Figure 5. (a) Correlation plots of the retrieval within the matching grid of CALIPSO nighttime and SAGE III/ISS product from 5 km to 30 km for 12 months of validation. The color bar represents the sample size. The black bins represent the mean values of each 10% quantile (0-10%, 10-20%...and 90-100%) of SAGE III/ISS 521 nm aerosol product and corresponding CALIPSO retrieval. The I-type bars indicate the standard error of each 10% quantile CALIPSO retrieval. (b) The relative uncertainty of one-degree CALIPSO extinction.

**Specific comments**

1. Lines 69-70: "The CALIPSO lidar is highly sensitive to cloud/aerosol layers with a lower bound of optical depth…". What is the lidar highly sensitive to? Presumably layer detection is meant, but the sentence does not say.

   ***Response:*** Thanks for your suggestion. This paragraph has been reorganized and this sentence has been deleted.

2. Line 70: "minimum detected extinction of 0.01 to 0.02 /km." This statement does not accurately represent the order of the level 2 CALIOP algorithms. The level 2 algorithms do not detect extinction. Layers are first detected using attenuated scattering ratios and then extinction is retrieved. Suggest rewording to clarify.

   ***Response***: Thanks for your suggestion. This paragraph has been reorganized and this sentence has been deleted. And the similar expressions of "detected extinction" elsewhere in the manuscript have been revised.

3. Line 93-94: Rather than just reference the Kar et al., 2019 paper to explain how clouds and aerosols are removed, it is recommended to add a sentence or two summarizing the removal procedure of that paper. Also, Kar et al., 2019 applies additional filters to remove undetected cloud layers beyond just using the VFM to

cloud-clear. Does the method for this manuscript do the same?

***Response***: Thanks for your suggestion. We have added an operation for residual cirrus cloud removal, which is illustrated in Section 2.1 now:

(2)  The clouds and aerosol layer detected by the SIBYL and the data below them were removed. We used a threshold value of 0.5 in the attenuated color ratio (the ratio of the TAB at 1064 and 532 nm) to remove undetected tenuous cirrus clouds, similar to the data screening method of the CALIPSO Level 3 Stratospheric Aerosol Profile product (Kar et al., 2019).

4.  Line 95: "The TAB is averaged at a vertical resolution of 300 m…" How is it possible to average the TAB to 300 m vertical resolution from 20.2 to 30.1 km when the range bins reported in the level 1 data product are at 180 m vertical resolution for that altitude region? Averaging two bins together in this region would yield 360 m, not 300 m. Furthermore, the TAB is already reported at 300 m vertical resolution from 30.1 – 40 km, so no average is required. Please clarify if the averaging used for this study considers the vertical resolution of the range bins in the level 1 data products.

***Response***: Sorry for the confusion. In the revised manuscript, we have added a description of the vertical resolution of the CALIPSO Level 1 data and explained that we reducing the vertical resolution to 300 m by linear interpolation in Section 2.1:

(3)  The vertical resolution of the CALIPSO Level 1B TAB profiles varies with the height of 30, 60, 120, and 300 m for −0.5–8, 8–20.2, 20.2–30.1, and 30.1–40 km, respectively. Referring to Kim et al. (2017), the TAB profiles are reduced to a vertical resolution of 300 m by linear interpolation to improve the SNR, followed by a vertical moving mean filtering (with a 5-point window) and horizontal averaging to 20 km to retrieve the extinction of undetected aerosol.

5.  Line 147. According to this line, the extinction retrieval yields a value of 0.01 /km. However, the text suggests the layer is smoke, so the lidar ratio being used is too low by 50/70. Therefore, this extinction value should be larger.

***Response***: Yes, this bias does still exist in the study and therefore in the revised manuscript we discuss this issue in Section 3.4:

Figure 7b illustrates a possibly missed smoke from a wildfire. Based on the SAGE-constrained lidar ratio (median 42.2 and 24.5 sr), we retrieved and see the undetected aerosol by CALIPSO Level 2 products, which connects with two strong

aerosol layers. The lidar ratio for the smoke in the CALIPSO Level 2 Version 4 product is 70±16 sr (Young et al., 2018), which is very different from the SAGE-constrained lidar ratio for the troposphere at this location. Theoretically, a larger lidar ratio will derive a larger extinction in the retrieval. This indicates that the undetected aerosol extinction should be larger if using the smoke lidar ratio of 70±16 sr. However, this bias cannot be avoided here because an automatic classification is impossible when we do not know the boundaries of those smoke aerosols. Therefore, we have to treat the stratospheric (or tropospheric) undetected aerosols as a whole and assign the same lidar ratio regardless of the aerosol type in this study. Although the retrieved extinction in Figure 7 is biased, it demonstrated the importance of retrieving high spatial-temporal resolution undetected aerosol extinction. A solution to reduce this bias is to develop a more effective layer detection and classification algorithm, and our team is already working on it (Mao et al., 2021).

6. Figure 3. According to pre-processing step (a), clouds and aerosols detected by CALIPSO are removed, along with the data beneath them. However, the purple boundaries in Fig. 3(c) shows that a smoke layer is detected and there is an extinction coefficient reported there. The text even quotes the extinction value on line 148 and panel (d) shows where these layers are detected. I thought that the backscatter was supposed to be removed where layers are reported. Why are they shown in this figure? They are not shown in Figure 7. This should be made clear somewhere which data is used in the retrieval shown in the extinction figure.

***Response***: Sorry for the confusion, In the revised manuscript we have redrawn this figure as follows:

[Figure]

Figure 7. (a) MODIS Terra true-color image in the daytime and the passing CALIPSO track (yellow line) at night. (b) Latitude–altitude aerosol extinction of the corresponding nighttime CALIPSO track, same as in Figure 4a. The purple and black boundary lines and extinction inside represent the detected aerosol and cloud layers provided by CALIPSO Level 2 products, respectively. (c) Aerosol subtypes in CALIPSO VFM product (N/A=not applicable, 1=marine, 2=dust, 3=polluted continental/smoke, 4=clean continental, 5=polluted dust, 6=elevated smoke, 7=dusty marine, 8=PSC aerosol, 9=volcanic ash, 10=sulfate/other). (d) Attenuated scattering ratio.

7. Figure 3 caption. "…additional mean filtering (3x3 window) to highlight the faint aerosol area." The premise of the paper is that averaging to 20 km x 300 m resolution is enough to highlight the faint aerosol. Why is additional averaging needed? Can these features still be discerned without this additional filtering? If not, then should the 3x3 window filtering be included as part of the methodology?.

*Response*: Sorry for the confusion. The simple $3\times3$ mean filtering was used in the mapping of the figure just for visual convenience.

8. Line 173: "…indicating a low bias in the CALIPSO retrieval." Some clarification should be added here because there could be two interpretations of this statement. (1) Because the CALIPSO level 2 layer detection did not capture these extinction values, there is a low bias in what CALIPSO reports. Or, (2) the retrieval of extinction from the CALIPSO products performed in this study has a low bias. Please clarify which condition this statement is addressing.

*Response:* We are very sorry for the confusion here. The related sentences have been rewritten as follows:

The nighttime CALIPSO undetected aerosol extinction and SAGE III/ISS 521 nm aerosol extinction show good agreement for the 12-month validation dataset (Figure 5a), with the average retrieved aerosol extinction (black line) closing to the 1:1 line. The correlation coefficients (R) and normalized root mean square error (NRMSE) are 0.66 and 100.6% based on the independent 12-month SAGE validation dataset, respectively.

9. Line 176: "…we can see that the retrieved aerosol extinction is much less than the detection limit (0.01 km$^{-1}$) of the CALIPSO Level 2 product". More precise language is requested here. The CALIPSO level 2 algorithms do not detect extinction, they detect layers and then retrieve extinction. This study addresses the extinction from aerosol layers below the layer detection limit of the level 2 feature

finder.

*Response*: Modified, as follows:

This indicates the retrieved extinction of undetected aerosol is much smaller than the low boundary of the detected aerosol extinction ($10^{-2}$ km$^{-1}$) from the CALIPSO Level 2 extinction product with a 40% uncertainty (Kacenelenbogen et al., 2011; Toth et al., 2018; Winker et al., 2013; Winker et al., 2009).

10. Lines 198 – 200: "Young et al. (2013) noted that the CALIPSO retrievals with SNR≤1 usually contain a positive bias. The SNR during daytime above 20 km is usually less than 1 for TAB at 20 km horizontal scale (Figure 6b), which leads to a significantly positive bias in the retrieval" It is not immediately clear how an SNR < 1 yields a positive bias. SNR speaks toward the (inverse of the) variability with respect to the average value, but not necessarily a bias. I would assume that a bias is more governed by calibration rather than noise. Or is it that the noise is not Gaussian? Please add information as to why a "significantly positive bias" is expected in the retrieval when SNR < 1.

*Response*: Additional explanations have been given in the manuscript as to why positive bias occurs when the SNR < 1, as follows:

The distribution of lidar signals received by photomultipliers is Neyman type-A (originally defined for a Poisson process) (Teich, 1981), thereby introducing a positive bias in the extinction retrieval calculation when the SNR is low. Also, Young et al. (2013) noted that the CALIPSO retrievals with SNR≤1 usually contain a positive bias. The SNR during daytime above 20 km is usually less than 1 for TAB at a 20 km horizontal scale (Figure 6b), leading to a significantly positive bias in the retrieval (Figure 6a), as noted by Young et al. (2013).

11. Line 210: It would be helpful to explain why the white areas of missing data occur between ± 15° in the level 3 panels of Figure 7 (because of the tropopause height).

*Response*: The missing data between ± 15° in the original manuscript is due to the higher tropospheric height at lower latitudes. In the revised manuscript, the height of the CALIPSO Level 3 product has been changed to 17 km to correspond to the aerosol enhanced area in the retrieved extinction scene map:

[Figure]

Figure 8. (a) and (b) are the stratospheric extinction distributions of CALIPSO Level 3 Stratospheric Aerosol Profile products at 17 km in June and August, respectively. (c) and (d) are the retrieved aerosol extinction scenes based on CALIPSO instantaneous data on June 10 and August 10, respectively, consistent with Figure 4a. The corresponding trajectories for the two scenes are shown as red lines in (a) and (b), and the corresponding aerosol subtypes are shown in (e) and (f), the same as in Figure 7c.

12. Section 3.3. Possible smoke from Siberian wildfires is not the only explanation for aerosol enhancement in the stratosphere during this time period. The June 2019 Raikoke volcano eruption also emitted a substantial amount of sulfate at northern latitudes. This should be discussed as part of the explanation and interpretation for aerosol enhancement in August 2019.

*Response*: Sorry for the confusion. We check the aerosol distribution at tropopause and stratosphere during the summer of 2019 based on CALIPSO L3 monthly-average data. The aerosol enhancements in August of 2019 are consistent with Raikoke volcanic eruption, as shown in Figure 8. Further, we compare aerosol at 17 km in August of 2018 and 2019 (Figure R1) to confirm these aerosol enhancements are unique in 2019. The Raikoke volcanic eruption has been added to the interpretation of stratospheric aerosol enhancement in the Northern

Hemisphere in August 2019 (Knepp et al., 2021; Kloss et al., 2021), as follows:

A significant amount of aerosol enhancement was observed in the stratosphere in August in the northern hemisphere (Figure 8b), possibly due to the eruption of the Raikoke Volcano in June 2019 (Knepp et al., 2021; Kloss et al., 2021).

[Figure]

Figure R1 (a) and (b) are the stratospheric extinction distributions of CALIPSO Level 3 Stratospheric Aerosol Profile products at 17 km in August 2018 and August 2019, respectively.

13. Line 225: "These faint aerosols propagate from 60N to near 10N…" The word "propagate" might be inaccurate for this discussion.

    *Response*: Thanks for your suggestion, we have removed the expression about "propagate", as follows:

    Following the onset of volcanic eruptions, strong stratospheric aerosol layers are found in the stratosphere between 50°N and 60°N that are classified as sulfate by the VFM (Figure 8f). As shown in the red dash box of Figure 8d, aerosol extinction enhancement (~0.005 km$^{-1}$) occurs around 17 km near 40°N to 5°N, which corresponds to the monthly average scale aerosol contamination in the stratosphere throughout the Northern Hemisphere in Figure 8b, but is not captured by CALIPSO Level 2 products (Figure 8f).

14. Lines 243 – 244: "The retrievable aerosol extinction greatly extends to 0.0001 km-1…" What is the relative error on these very low extinction values?

    *Response*: A discussion of the relative uncertainty in the retrieved extinction has been added to the revised manuscript, as described in the response of Major Comment 3 and in the following:

    The averaged black line in Figure 5b show the mean relative uncertainties of CALIPSO, specifically ~35% and ~125% for the retrieved extinction of 10$^{-3}$ and 10$^{-4}$ km$^{-1}$, respectively.

15. Lines 244 – 246: "The comparison is unavailable at low altitudes, but the retrieval should be more reliable (i.e., in the troposphere) because the SNR is higher." The improvement in SNR is only part of the story. A far more substantial factor that will cause larger errors in the troposphere is the choice of lidar ratio, which can range from 20 – 70 sr. This can cause the biases up to a factor of three when the wrong lidar ratio is used. It is important to include a discussion on how the choice of lidar ratios for this analysis impacts comparisons with SAGE retrievals.

    *Response*: You are right. We have removed the comparison on SNR of SAGE III/ISS and CALIPSO and added the discussion of the effect of lidar ratio selection on retrieved extinction, as described in the response of Major Comment 2.

16. Lines 247 – 248. Same question as before, how does low SNR yield a positive bias? More should be added here to summarize why this is true.

    *Response*: Additional explanations have been given in the manuscript as to why positive bias occurs when the SNR < 1, as follows:

    The distribution of lidar signals received by photomultipliers is Neyman type-A (originally defined for a Poisson process) (Teich, 1981), thereby introducing a positive bias in the extinction retrieval calculation when the SNR is low. Also, Young et al. (2013) noted that the CALIPSO retrievals with SNR≤1 usually contain a positive bias. The SNR during daytime above 20 km is usually less than 1 for TAB at a 20 km horizontal scale (Figure 6b), leading to a significantly positive bias in the retrieval (Figure 6a), as noted by Young et al. (2013).

17. Lines 249 – 252. A couple of points about conclusion item (3).

    First, the stratospheric aerosol enhancement for this time period includes contributions from the Raikoke volcanic plume in addition to (possible) smoke from Siberian wildfires. This should be included with the discussion of sources for this example. There is some discussion in the literature about the contribution of these aerosol types in August 2019.

    Second, this sentence can easily be interpreted as an over-generalization, "our retrieval shows that these faint aerosols even propagate to near 10°N, which is much beyond the detecting range of the CALIPSO L2 products (50° N and 60° N)." I believe this sentence is a summary of the single level 2 granule evaluated in Figure 7 where the level 2 algorithms did not detect a large extent of the stratospheric aerosol enhancement from 10N to 50N, Fig 7(d). For this specific case, the aerosol was not detected by CALIOP level 2. However, the sentence is written as though this is a general result: faint aerosols following the 2019 Siberian fires (and

Raikoke eruption) are not detected as far south as 10N by CALIOP level 2 retrievals. This cannot be concluded based on the one granule examined. To make the possibility of misinterpretation more probable, Figure 7(b) shows nothing reported in the CALIOP level 3 stratospheric aerosol product from about 15 N/S. This is merely because the tropopause is above 15.2 km at those latitudes, but a reader could easily read this sentence and look at Figure 7(b) and conclude that CALIOP level 2 missed detecting all of that aerosol during August 2019. It is unlikely that CALIOP level 2 did not detect all of this aerosol, and even if so, it was not proven in the manuscript. I recommend rephrasing conclusion item (3) to be more specific on the evidence for the conclusion being made.

***Response***: Thank you for your comment. Based on the new methods, results and discussions, we rewrote the conclusions 
[revised manuscript text omitted]

---

## Editor Decision (ED1)

**Manuscript ID:** acp-2022-56

**Title:** Retrieving Instantaneous Extinction of Aerosol Undetected by CALIPSO Layer Detection Algorithm

**Comments to the author from the Editor**:

Dear Authors,

In general, you have successfully implemented the recommended corrections/revisions from the referees. However, you have also made changes to the text in addition to what was suggested by the referees. I have some suggested changes to these additional modifications and a few additional comments as follows:

1. Lines 32-35. Suggest rewording "However, aerosols still represent a major uncertainty in global climate change and energy balance with a low scientific understanding (Lee et al., 2016; Watson-Parris et al., 2020), which is partly attributed to not enough observations for three-dimensional (3D) aerosol distribution characteristics." to something along the lines of:
   "However, aerosols still represent a major uncertainty in global climate change and energy balance with a low scientific understanding (Lee et al., 2016; Watson-Parris et al., 2020), which is partly attributed to insufficient observations to accurately characterize the three-dimensional (3D) aerosol distribution."

2. Lines 43-47. The revised wording of these two sentences is confusing to me. The original sentences read:
   "However, these faint aerosols are usually extremely optically thin to be detected by the CALIPSO layer detection algorithm with a minimum 0.05 threshold of column aerosol optical depth (AOD) . A previous study indicated the retrieved AODs of aerosols undetected by the CALIPSO layer detection algorithm can reach 0.03–0.05, which accounts for approximately 20% of the total AOD and are very important for climatology (Toth et al., 2018; Smirnov et al., 2011; Levy et al., 2013)."

   The new sentences read:
   "However, these faint aerosols are usually extremely optically thin to be detected by the CALIPSO layer detection algorithm. Previous studies indicated the aerosols undetected and retrived by the CALIPSO generally have the aerosol optical depth (AOD) of 0.03–0.05, which accounts for approximately 20% of the total AOD and are very important for climatology (Toth et al., 2018; Smirnov et al., 2011; Levy et al., 2013)."

   First of all, retrieved is misspelled in L. 45. But the inclusion of the word "retrieved" now implies that the aerosols are undetected but retrieved by the CALIPSO (algorithm). I'm not sure what that actually means. Is that what you intend to say? Otherwise, you

should add the word "not" before retrieved: "undetected and not retrieved by the CALIPSO algorithm…"

I think the original wording is clear and see no reason to reword these sentences.

3. Line 110. "Due to this bias is negligible at 450 and 755 nm."
   What is negligible? The low bias? If yes, then the sentence should read something like: "This low bias is negligible in the 450 and 755 nm aerosol channels."

4. Equation 8.   This is an estimate of the uncertainty in extinction due to the two sources on the right hand side of the equation. It is odd that the first term on the right is the sum of the squares of errors (this is typical error propagation) but the second term is the square of the sum of the errors. Is this a typo?

---

## Author Response (AR2)

**Manuscript ID:** acp-2022-56

**Title:** Retrieving Instantaneous Extinction of Aerosol Undetected by CALIPSO Layer Detection Algorithm

**Comments to the author**:

Dear Authors

Please implement the minor corrections/revisions as recommended by Anonymous Referee #4 in the most recent Open Report #2. Once these are completed, the paper will be accepted for publication. Thank you.

**Specific comments**

1.  Line 73. The level 2 products do not explicitly report chemical parameters. Recommend removing "chemical"

    **RE:** Removed.

2.  Table 1 and lines 140-143. The molecular number density, ozone number density, and tropopause height are provided by the MERRA-2 model. Adding this information into the manuscript will help avoid giving the impression that CALIOP retrieves these values.

    **RE:** The related sentence has been added at the caption of Table 1, as follows 'The tropopause height, molecular number density and ozone number density are provided by the Global Modeling and Assimilation Office.'

3.  Line 107. I do not think the Knepp et al. 2021 is an appropriate reference because it is still in discussion. Please consult the ACP editorial staff for their standards about this. I believe the Knepp 2021 paper references another paper when describing the extinction to correction. If so, consider referencing that other paper instead since it has fully completed the peer review process. Similarly, the Knepp 2021 reference is used in this manuscript to describe the Raikoke event (line 281). There are other papers that have been published describing the eruption observations that may be used instead if you do not want to reference a paper in

discussion. If the ACP editors are ok with referencing a paper in discussion, then I withdraw my comment.

**RE:** The Knepp et al. 2021 have been replaced to the published papers, as follows:

1) Wang, H. J. R., et al. Validation of SAGE III/ISS solar occultation ozone products with correlative satellite and ground based measurements. Journal of Geophysical Research: Atmospheres, 125, e2020JD032430. https://doi.org/10.1029/2020JD032430, 2020.

2) de Leeuw, J., et al. The 2019 Raikoke volcanic eruption – Part 1: Dispersion model simulations and satellite retrievals of volcanic sulfur dioxide, Atmos. Chem. Phys., 21, 10851–10879, https://doi.org/10.5194/acp-21-10851-2021, 2021.

4. Lines 229-232 "…is the 20 km aerosol extinction of CALIPSO", "is the uncertainty for one-degree aerosol extinction of CALIPSO", "…or the CALIPSO retrieval". It is not clear with the expressions if they refer to the extinction and uncertainties reported in the official CALIPSO products or if they refer to the values calculated in this study. Can this be made specific?

**RE:** Sorry for the confusion. The related content has been revised as follows:

'The CALIPSO extinction in Figure 5a comes from the averaged extinction profiles over the one-degree matched range with SAGE (Figure 1a). They are equal to the average of the five 20-km extinction profiles over the matched range. Therefore, considering the systematic error of the lidar ratio in Eq. 7, we calculate the uncertainty of averaged extinction within one-degree range according to the following equation:

$$(\Delta\alpha_{1\circ})^2 = \sum_{i=1}^{n}(\frac{1}{n} \times \left(\frac{\Delta\beta_p}{\beta_p}\right) \times \alpha_{20\ km})^2 + \left(\sum_{i=1}^{n}(\frac{1}{n} \times \left(\frac{\Delta S_p}{S_p}\right) \times \alpha_{20\ km}\right)^2, \qquad (8)$$

5. Line 284. Raikoke misspelled.

**RE:** Revised

6. Lines 294-296. "The results show that the aerosol subtype in this region is sulfate,

which supports that the aerosol enhancement is more 295 likely to be from the eruption of the Raikoke Volcano." Actually, there is some literature that suggests Siberian wildfire smoke may be contributing to stratospheric aerosol in the region where figure 8d passes through (Ansmann et al., 2021). It is worth considering adding an acknowledgment of the possibility in this sentence.

**RE:** We have added an acknowledgment about the possible effect of Siberian wildfire on the enhanced stratospheric aerosols, as follows:

'Additionally, the previous study indicates the Siberian wildfire smoke possibly contribute to the enhanced stratospheric aerosols in 2019 by the combining CALIPSO observation and backward trajectories model (Ansmann et al., 2021).'

Ansmann. A., K. Ohneiser, A. Chudnovsky, H. Baars and R. Engelmann, 2021: "CALIPSO Aerosol-Typing Scheme Misclassified Stratospheric Fire Smoke: Case Study From the 2019 Siberian Wildfire Season", Front. Environ. Sci., 9:769852, https://doi.org/10.3389/fenvs.2021.769852.

**References**

Ansmann, A., Ohneiser, K., Chudnovsky, A., Baars, H., and Engelmann, R.: CALIPSO Aerosol-Typing Scheme Misclassified Stratospheric Fire Smoke: Case Study From the 2019 Siberian Wildfire Season, Frontiers in Environmental Science, 9. doi:10.3389/fenvs.2021.769852, 2021.

---

## Author Response (AR3)

**Manuscript ID:** acp-2022-56

**Title:** Retrieving Instantaneous Extinction of Aerosol Undetected by CALIPSO Layer Detection Algorithm

**Comments to the author:**

Dear Authors,

In general, you have successfully implemented the recommended corrections/revisions from the referees. However, you have also made changes to the text in addition to what was suggested by the referees. I have some suggested changes to these additional modifications and a few additional comments as follows:

1. Lines 32-35. Suggest rewording "However, aerosols still represent a major uncertainty in global climate change and energy balance with a low scientific understanding (Lee et al., 2016; Watson-Parris et al., 2020), which is partly attributed to not enough observations for three-dimensional (3D) aerosol distribution characteristics." to something along the lines of: "However, aerosols still represent a major uncertainty in global climate change and energy balance with a low scientific understanding (Lee et al., 2016; Watson-Parris et al., 2020), which is partly attributed to insufficient observations to accurately characterize the three-dimensional (3D) aerosol distribution."

   **RE:** Thank you for your suggestion. Revised.

2. Lines 43-47. The revised wording of these two sentences is confusing to me. The original sentences read: "However, these faint aerosols are usually extremely optically thin to be detected by the CALIPSO layer detection algorithm with a minimum 0.05 threshold of column aerosol optical depth (AOD). A previous study indicated the retrieved AODs of aerosols undetected by the CALIPSO layer detection algorithm can reach 0.03–0.05, which accounts for approximately 20% of the total AOD and are very important for climatology (Toth et al., 2018; Smirnov et al., 2011; Levy et al., 2013)."

   The new sentences read: "However, these faint aerosols are usually extremely

optically thin to be detected by the CALIPSO layer detection algorithm. Previous studies indicated the aerosols undetected and retrived by the CALIPSO generally have the aerosol optical depth (AOD) of 0.03–0.05, which accounts for approximately 20% of the total AOD and are very important for climatology (Toth et al., 2018; Smirnov et al., 2011; Levy et al., 2013)."

First of all, retrieved is misspelled in L. 45. But the inclusion of the word "retrieved" now implies that the aerosols are undetected but retrieved by the CALIPSO (algorithm). I'm not sure what that actually means. Is that what you intend to say? Otherwise, you should add the word "not" before retrieved: "undetected and not retrieved by the CALIPSO algorithm…" I think the original wording is clear and see no reason to reword these sentences.

**RE:** Sorry for the confusion. We change the related sentences back to the original version, as follows: 'However, these faint aerosols are usually extremely optically thin to be detected by the CALIPSO layer detection algorithm. A previous study indicated the retrieved AODs of aerosols undetected by the CALIPSO layer detection algorithm can reach 0.03–0.05 (Toth et al., 2018), which accounts for approximately 20% of the total AOD and are very important for climatology (Toth et al., 2018; Smirnov et al., 2011; Levy et al., 2013).'

3. Line 110. "Due to this bias is negligible at 450 and 755 nm." What is negligible? The low bias? If yes, then the sentence should read something like: "This low bias is negligible in the 450 and 755 nm aerosol channels."

   **RE:** You are right. Revised as your suggestion.

4. Equation 8. This is an estimate of the uncertainty in extinction due to the two sources on the right hand side of the equation. It is odd that the first term on the right is the sum of the squares of errors (this is typical error propagation) but the second term is the square of the sum of the errors. Is this a typo?

   **RE:** Sorry for the confusion. The first term on the right indicates the random error of averaged extinction at one-degree range. It is equal to the sum of the squares of

errors based on the five related extinction values at 20-km horizontal resolution. However, the second term indicates the systematic error from lidar ratio ($S_p$) uncertainty, and is equal to the square of averaged systematic errors of 20-km extinction values. Based on the error propagation principle, the systematic error should not decrease with the average scale.

We slightly revised Equation 8 to make it clearer, and add the related description about the uncertainty with the conderation of systematic components.

$$(\Delta\alpha_{1^\circ})^2 = \sum_{i=1}^{n}\left(\frac{1}{n} \times \frac{\Delta\beta_{p,i}}{\beta_{p,i}} \times \alpha_{20\ km,i}\right)^2 + \left[\sum_{i=1}^{n}\left(\frac{1}{n} \times \frac{\Delta S_{p,i}}{S_{p,i}} \times \alpha_{20\ km,i}\right)\right]^2, \tag{8}$$

where $n$ represents the number of CALIPSO 20 km profiles ($i$ =1, 2, …, n) in the matching range, $\alpha_{20\ km}$ is the 20 km aerosol extinction of CALIPSO, and $\Delta\alpha_{1^\circ}$ is the uncertainty for one-degree aerosol extinction of CALIPSO. The first term on the right indicates the random error, and is equal to the sum of the squares of errors based on the five related 20-km extinction values. The second term indicates the systematic error from the lidar ratio, and should not decrease with the average scale based on the error propagation principle.